# The RNA-binding protein AKAP8 suppresses tumor metastasis by antagonizing EMT-associated alternative splicing

Xiaohui Hu[1,2,4], Samuel E. Harvey[1,2,4], Rong Zheng[1,2], Jingyi Lyu[1,2], Caitlin L. Grzeskowiak[2], Emily Powell[3], Helen Piwnica-Worms [3], Kenneth L. Scott[2] & Chonghui Cheng [1,2]*

Alternative splicing has been shown to causally contribute to the epithelial–mesenchymal transition (EMT) and tumor metastasis. However, the scope of splicing factors that govern alternative splicing in these processes remains largely unexplored. Here we report the identification of A-Kinase Anchor Protein (AKAP8) as a splicing regulatory factor that impedes EMT and breast cancer metastasis. AKAP8 not only is capable of inhibiting splicing activity of the EMT-promoting splicing regulator hnRNPM through protein–protein interaction, it also directly binds to RNA and alters splicing outcomes. Genome-wide analysis shows that AKAP8 promotes an epithelial cell state splicing program. Experimental manipulation of an AKAP8 splicing target CLSTN1 revealed that splice isoform switching of CLSTN1 is crucial for EMT. Moreover, AKAP8 expression and the alternative splicing of CLSTN1 predict breast cancer patient survival. Together, our work demonstrates the essentiality of RNA metabolism that impinges on metastatic breast cancer.

[1] Lester & Sue Smith Breast Center, Baylor College of Medicine, Houston, TX 77030, USA. [2] Department of Molecular and Human Genetics, Baylor College of Medicine, Houston, TX 77030, USA. [3] Department of Experimental Radiation Oncology, The University of Texas MD Anderson Cancer Center, Houston, TX 77030, USA. [4] These authors contributed equally: Xiaohui Hu, Samuel E. Harvey. *email: chonghui.cheng@bcm.edu

Tumor metastasis is the most lethal attribute of breast cancer. One of the key mechanisms that facilitates cancer metastasis is the abnormal activation of a developmental process termed epithelial–mesenchymal transition (EMT)[1–3]. Aberrant activation of EMT enables primary epithelial cancer cells to acquire advantageous mesenchymal properties, including invasion and drug resistance, ultimately allowing the survival of cancer cells within the circulatory system and subsequent colonization of distant organs[4–6]. Whereas several transcription factors, such as Twist, Snail, and Zeb1/2, and signaling pathways, including TGF-β, have been characterized as potent inducers of EMT[6,7], growing evidence has suggested that alternative RNA splicing acts as a critical layer of regulation impinging on EMT[8–11].

Alternative RNA splicing is a fundamental mechanism of post-transcriptional gene regulation. With 95% of human multi-exon genes expressing more than one splice isoform, alternative splicing contributes to the diversity and complexity of the human proteome, and thus organ development and tissue identity[12–14]. The regulation of alternative splicing relies on the precise binding of splicing factors to the RNA consensus motifs located in variable exons or their adjacent introns. Therefore, mutations in either splicing factors or RNA motifs that perturb splicing factor binding may result in developmental abnormalities and diseases[15,16]. Although important observations connecting splicing machinery and diseases are accumulating, our understanding of the mechanisms and functions of splicing regulation that impinges on diseases is still in its infancy.

The functional connection of alternative splicing to EMT and cancer metastasis was established through the study of the *CD44* gene, which is alternatively spliced to generate two families of proteins, known as CD44v and CD44s. Following our initial discovery that CD44 isoform switching is essential for EMT[8], other studies have also reported that epithelial cells that predominantly express CD44v demand an isoform switch to CD44s in order for cells to undergo EMT and for cancer cells to metastasize[17–26]. In addition to CD44, a handful of additional alternative splicing events has subsequently been reported to play a functional role in EMT[27–30]. EMT-associated splicing events are controlled by splicing factors and, to a large extent, these splicing factors act in a combinatorial manner to influence splicing[9,10,31,32]. In the case of CD44 alternative splicing, the heterogeneous nuclear ribonucleoprotein M (hnRNPM) promotes the production of CD44s by binding to *CD44* intronic splicing motifs, resulting in an EMT phenotype and enhanced metastasis[10]. The splicing activity of hnRNPM is partially restricted by an epithelial-specific splicing factor ESRP1 through competitive binding to the same RNA motifs, thus tightly controlling the switch of CD44 splice isoforms and transition of cell states during EMT[9,10]. In addition to this mode of direct competition through binding to RNA substrates, it is conceivable that hnRNPM-interacting splicing factors could also influence hnRNPM's activity and thus its function in promoting EMT. In fact, several splicing factors were found to form a complex with hnRNPM[31,33,34], but the functional consequences in EMT and cancer metastasis remained unexplored.

In this study, we report the identification of the A kinase anchoring protein 8 (AKAP8) as an RNA-binding protein that inhibits EMT and breast cancer metastasis through the regulation of alternative splicing. AKAP8 interacts with hnRNPM and precludes the activity of hnRNPM to stimulate exon skipping of *CD44*. Moreover, AKAP8 is capable of directly binding to RNA and modulating alternative splicing events. Functionally, AKAP8 is required to maintain epithelial-specific alternative splicing patterns. Cells with loss of AKAP8 show accelerated EMT and enhanced breast cancer metastatic potential. We demonstrate that both AKAP8 and its splicing target CLSTN1 accurately predict patient survival. These results identify the splicing factor AKAP8 as a suppressor of EMT and metastatic cancer and shed lights on the mechanisms of EMT and tumor metastasis that are regulated at the level of alternative RNA splicing.

## Results

**AKAP8 interacts with hnRNPM and correlates with patient survival.** To identify splicing regulatory proteins that regulate EMT and metastasis, we used hnRNPM as a bait to determine hnRNPM-interacting splicing factors. As hnRNPM stimulates mesenchymal-associated splicing and promotes cancer metastasis, we reasoned that these hnRNPM-interacting splicing factors likely affect EMT and tumor metastasis by synergizing with or antagonizing hnRNPM's activity. As depicted in Fig. 1a, we applied a BioID technology that utilizes a biotin-ligase fused to hnRNPM to capture hnRNPM-interacting proteins that are ligated with biotin in live cells (Supplementary Fig. 1a). Mass spectrometry analysis successfully revealed several previously reported hnRNPM-interacting splicing factors, including RBFOX, SFPQ, and PTBP1[31,35,36]. From the top 50 hnRNPM-interacting proteins, we selected 29 splicing factors (See Methods for details) and performed a *CD44v8* splicing minigene reporter assay (Supplementary Fig. 1b and Supplemental Data 1). After co-transfecting each of the 29 open-reading frame (ORFs) with the *CD44v8* minigene reporter to 293FT cells, we analyzed the levels of *CD44v8* splicing, depicted by the ratios of inclusion to skipping. Several splicing factors showed notable effects, i.e., greater than twofold upregulation and 2.5-fold downregulation of the ratios (Fig. 1b). Among them, PTBP1, AKAP8, and hnRNPF promoted *CD44v8* inclusion, and RBM10, RBMX, and hnRNPR promoted exon skipping. Immunoprecipitation validation showed that, except for PTBP1, the remaining five splicing factors interact with hnRNPM in an RNA-independent manner, and some of them showed even stronger protein interactions in the absence of RNA (Fig. 1c). Among the five splicing factors, hnRNPF was previously reported to stimulate *CD44v8* inclusion and inhibit EMT[37].

By examining the correlation between the above identified splicing factors and important clinical outcomes, we found that AKAP8 has the most significant correlation with metastasis and patient survival (Fig. 1d). AKAP8 expression positively correlates with distal metastasis-free survival in a cohort of 327 published breast cancer samples analyzed by microarray[38]. The positive correlation of AKAP8 expression and metastasis-free survival is congruent with our experimental findings that AKAP8 promotes *CD44v8* inclusion and inhibits CD44s production, the isoform that promotes EMT and tumor metastasis[8,26,39]. Further analysis of the METABRIC breast cancer data set showed that AKAP8 expression positively correlates with overall survival (Fig. 1e), most significantly in Luminal A, Luminal B, and HER2 + subtypes (Supplementary Fig. 1c–g). Analysis of AKAP8 expression levels in different subtypes of breast cancer revealed that AKAP8 expression is highest in the Luminal A subtype, the least aggressive breast cancer subtype that is epithelial in nature, and AKAP8 expression is significantly lower in the Claudin low and basal subtypes, which are more aggressive and mesenchymal (Fig. 1f and Supplementary Fig. 1h). Similarly, AKAP8 is highly expressed in the ER-positive breast tumors compared with the ER-negative breast tumors (Supplementary Fig. 1i). These results associate AKAP8 with an epithelial phenotype in breast cancer and show that loss of AKAP8 is a characteristic of poor survival, prompting us to explore the mechanistic roles of AKAP8 using in vitro models of EMT and in vivo models of breast cancer metastasis.

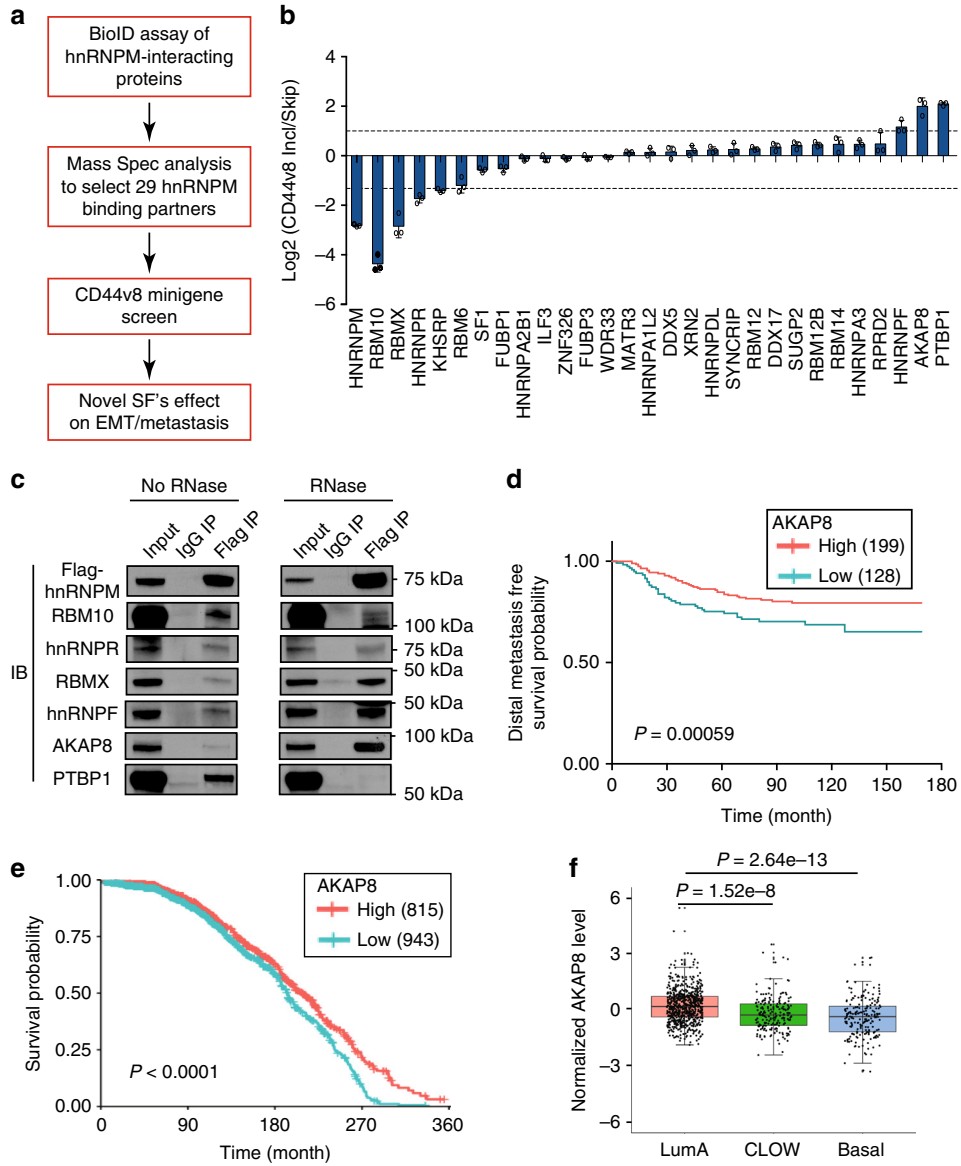

**Fig. 1 Functional screening to identify AKAP8 as an hnRNPM-interacting protein. a** A flow chart showing the experimental approaches to identify hnRNPM-interacting proteins. **b** qRT-PCR analysis of the *CD44v8* splicing reporter minigene screening for the candidate splicing factors. Data were plotted as the Log2 transformed v8 exon inclusion versus skipping with mean ± s.d, $n = 3$. Incl: Inclusion. **c** Western blot analysis showing the interactions between hnRNPM and its candidate interacting proteins. A Flag-tagged hnRNPM cDNA was transfected into the 293 cells and immunoprecipitated with a Flag antibody with or without RNase treatment. Antibodies recognizing specific candidates were used for western blot analysis. **d** Kaplan–Meier plot analysis of breast cancer patient distal metastasis-free survival (GSE20685, $n = 237$) showing that higher levels of AKAP8 expression predict lower metastatic potential. *P* value was calculated by log-rank test. **e** Kaplan–Meier plot analysis of the METABRIC breast cancer data set ($n = 1758$) showing that higher expression of AKAP8 shows better patient survival probability. **f** Box and whiskers plots with jitters representing distribution of AKAP8 mRNA expression levels in luminal A (LumA), claudin low (CLOW), and basal (Basal) breast cancers patients from the breast cancer METABRIC data set. The line within each box represents the median. Upper and lower edges of each box represent 75th and 25th percentile, respectively. The whiskers represent the maximum and minimum values within 1.5× the interquartile range. *P* values were calculated by two sample *z* test in **e**, **f**. Source data are provided as a Source Data file.

**Knockdown of AKAP8 promotes a mesenchymal phenotype.** The above observations suggest that AKAP8 protects an epithelial state, so we sought to determine whether knockdown of AKAP8 accelerates EMT. We utilized a tamoxifen (TAM)-inducible EMT system where human mammary epithelial cells were engineered to ectopically express the transcription factor Twist fused to ER (HMLE/Twist-ER, Ref. [8]). We depleted AKAP8 in HMLE/Twist-ER cells by two independent shRNAs and observed a marked knockdown of the AKAP8 protein (Fig. 2a, Supplementary Fig. 2a, b). With TAM induction, both the control and AKAP8

shRNA-expressing cells underwent morphological changes with gradual loss of the cobble-stone-like epithelial clusters and gain of expression of mesenchymal markers (Fig. 2a, b). The AKAP8 shRNA-expressing cells showed a more rapid transition to the mesenchymal phenotype compared to controls. At Day 12 of TAM induction, AKAP8 knockdown cells showed a more drastic reduction of epithelial markers E-cadherin and γ-catenin and a more pronounced increase in the mesenchymal marker N-cadherin (Fig. 2a). Although control cells were still loosely packed as clusters, the AKAP8 knockdown cells were fully transited to

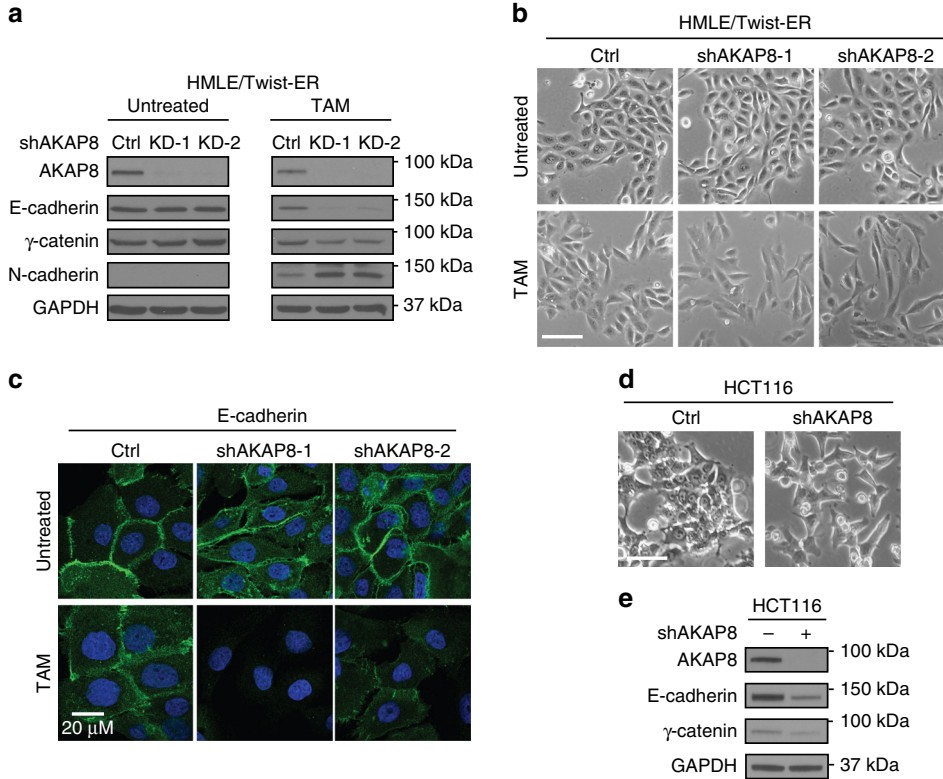

**Fig. 2 Depletion of AKAP8 promotes an EMT phenotype. a** Western blot analysis of EMT markers using lysates from HMLE/Twist-ER cells expressing control shRNA (Ctrl) or AKAP8 shRNAs (KD-1, KD-2). Lysates were collected before (untreated) and after 12 days of tamoxifen (TAM) induction. **b** Phase-contrast images (×10) displaying cell morphology differences between control (Ctrl) and AKAP8-silenced HMLE/Twist-ER cells before (untreated) and after 12 days of TAM induction. White line represents scale bar at 100 μm. **c** Immunofluorescence images (×40) showing the loss of E-cadherin at cell junction 12 days after TAM treatment in the AKAP8 knockdown cells. Green: E-cadherin, Blue: DAPI. **d** Phase-contrast images of HCT116 cells showing cell morphology changes upon AKAP8 knockdown. White line represents scale bar at 100 μm. **e** Western blot analysis of the epithelial markers in HCT116 cells expressing control or AKAP8 shRNA. Source data are provided as a Source Data file.

spindle-shaped mesenchymal cells (Fig. 2b) and nearly completely lost the adherens junction protein E-cadherin at cell junctions (Fig. 2c). These results reveal that AKAP8 knockdown accelerates EMT. To generalize the findings, we examined the effect of AKAP8 knockdown in a different epithelial cell line, HCT116 colon cancer cells, and observed that knockdown of AKAP8 alone is sufficient to induce a loss of epithelial characteristics, as demonstrated by the gain of elongated spindle-like mesenchymal morphology and the decrease in expression of epithelial markers (Fig. 2d, e). Re-expressing the AKAP8 cDNA in the AKAP8 knockdown cells restored the decreased expression of epithelial markers (Supplementary Fig. 2c). Together, these results indicate that knockdown of AKAP8 promotes EMT.

**AKAP8 inhibits breast cancer lung metastasis**. As EMT is essential for tumor metastasis, our finding that AKAP8 inhibits EMT prompted us to determine whether AKAP8 inhibits breast cancer metastasis. We used a patient-derived xenograft (PDX) model HIM3 where the cells were derived from a patient triple-negative breast tumor[40]. HIM3 maintains an epithelial phenotype and expresses high levels of E-cadherin and AKAP8 (Fig. 3a). Consistent with our observations shown in Fig. 2, knocking down AKAP8 in the HIM3 cells resulted in a decrease in epithelial markers E-cadherin, γ-catenin, and Occludin. To capture the effect of AKAP8 on tumor metastasis, we inoculated NSG mice with $1 \times 10^5$ control or AKAP8 knockdown cells through tail vein injection and measured the potential of lung metastasis (Fig. 3b). AKAP8 knockdown with two different shRNAs significantly

increased BLI signals compared with control, indicating the enhanced ability of metastatic tumor formation in mice in response to AKAP8 depletion (Fig. 3c, d). Our Hematoxylin and Eosin (H&E) analysis support this observation, showing a significant increase in the area of metastatic nodules in the lungs of mice that were injected with cells expressing AKAP8 shRNAs (Fig. 3e, f). We also noticed that the AKAP8 KD-2 cells showed greater metastasis ability than the KD-1 cells, although the KD-2 shRNA was less potent in promoting the EMT phenotype than the KD-1 shRNA. This observation could be in support of the growing evidence that cells in a hybrid epithelial/mesenchymal state tend to have a greater advantage in establishing metastatic lesions[41–44]. Because the AKAP8 shRNA-expressing HIM3 cells did not show a proliferation advantage compared with control HIM3 cells (Supplementary Fig. 3a), these results demonstrate that AKAP8 silencing promotes breast cancer metastasis to the lung.

As a complementary approach, we ectopically expressed AKAP8 in a lung metastatic breast cancer cell line LM2, a derivative of MDA-MB-231 cells, and examined whether forced expression of AKAP8 inhibits lung metastasis. Tetracycline-induced expression of AKAP8 in LM2 cells promoted the expression of epithelial marker and inhibited the levels of mesenchymal markers (Supplementary Fig. 3b, c). These cells also showed retardation of cell migration in vitro (Supplementary Fig. 3d, e). Interestingly, tail vein injection of the AKAP8-expressing LM2 cells in mice resulted in a drastic decrease in metastatic nodule formation as indicated by the luminescent signals and the H&E stains of lung sections (Fig. 3g-j). Taken

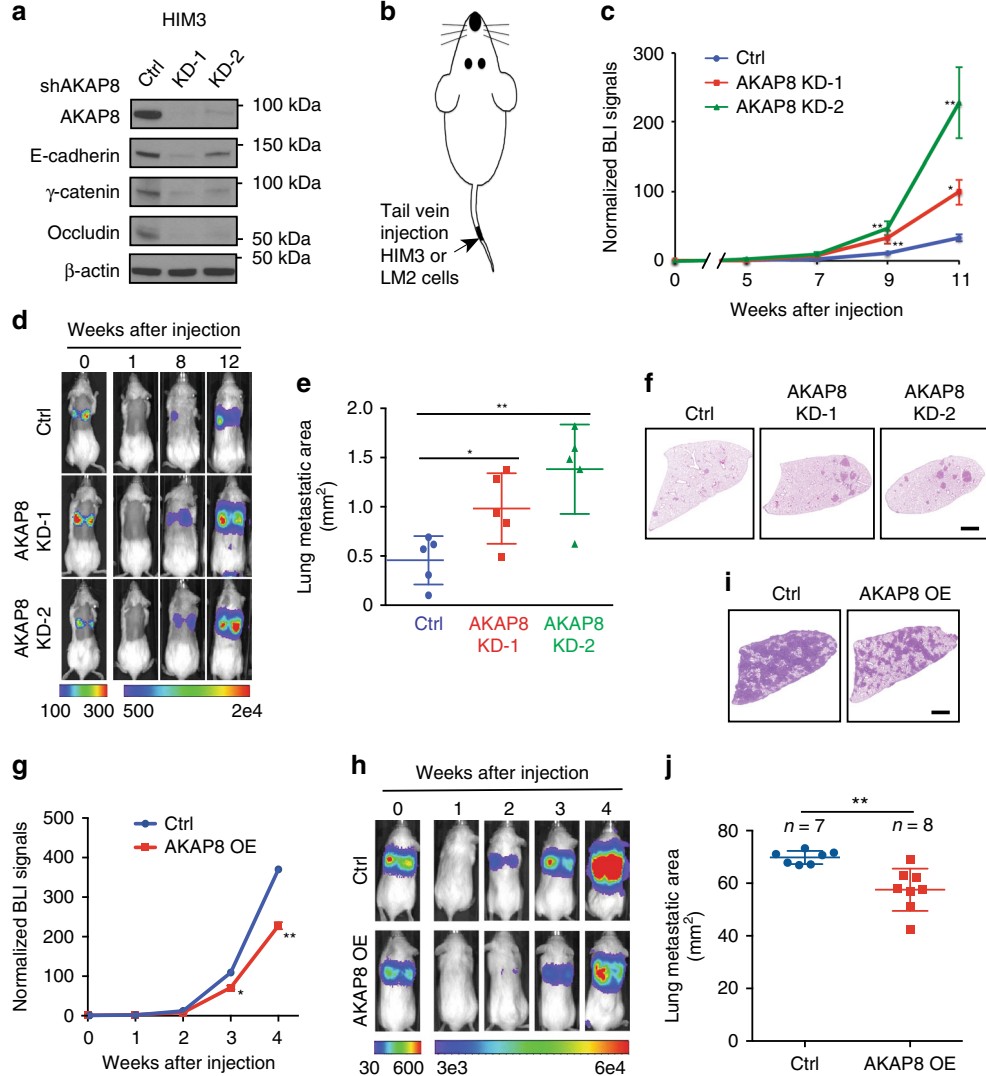

**Fig. 3 AKAP8 suppresses breast cancer metastasis. a** Western blot analysis showing the decreased expression of epithelial markers, E-cadherin, γ-catenin, and Occludin in HIM3 cells expressing control shRNA (Ctrl) and AKAP8 shRNAs (KD-1, KD-2). **b** Schematic of a xenograft model to measure breast cancer metastasis using the HIM3 PDX cells and the LM2 breasts cancer metastatic cells. **c–f** Tail vein injection of the HIM3 PDX tumor cells showing that silencing AKAP8 promotes metastatic tumor growth. Normalized bioluminescent imaging (BLI) signals **c** and representative BLI images **d** were shown at indicated time points. The areas of lung metastasis nodules were quantified by image J **e** and representative lung H&E stains **f** were displayed N = 5. **e** The middle line indicates mean of lung metastatic area, the top and bottom lines represent s.d. In **f**, black line represents scale bar at 2 mm. **g–i** Tail vein injection of LM2 cells showing that ectopic expression of empty control (Ctrl) and AKAP8 (AKAP8 OE) inhibits metastatic tumor growth. Normalized BLI signals **g**, representative BLI images **h**, lung H&E stains **i**, and quantifications of total lung metastasis areas **j** were shown (N > 7). **i** black line represents scale bar at 2 mm. **j** The middle line indicates mean of lung metastatic area, the top and bottom lines represent s.d. All Error bars indicate s.e.m. (*) P < 0.05; (**) P < 0.01. N > 7. P values were tested by Student's t test, two-tailed in **c**, **e**, **g**, **j**. Source data are provided as a Source Data file.

together, these gain-and-loss of function analyses of AKAP8 demonstrate that AKAP8 prevents breast cancer metastasis to the lung.

**AKAP8 antagonizes the splicing activity of hnRNPM.** Having established the function of AKAP8 in inhibiting tumor metastasis, we next sought to determine its underlying mechanisms. As hnRNPM promotes EMT and breast cancer metastasis by stimulating *CD44* exon skipping[10], we hypothesized that AKAP8 antagonizes the splicing activity of hnRNPM. To test this, we co-transfected the *CD44v8* splicing minigene construct (Supplementary Fig. 1b) that contains *CD44* variable exon 8 flanked by introns and two constitutive exons with the AKAP8 cDNA into 293FT cells. We found that AKAP8 promotes exon inclusion of

the v8 exon in a dose-dependent manner (Supplementary Fig. 4a) and that the effect of hnRNPM on inhibiting *CD44v8* exon inclusion was dampened in the presence of AKAP8 (Fig. 4a and supplementary Fig. 4b). These results suggest that AKAP8 antagonizes hnRNPM's activity and were recapitulated using another splicing minigene reporter that contains the *CD44* variable exon 5 (Fig. 4b, c and Supplementary Fig. 4c). We further found that, when AKAP8 is silenced, hnRNPM elicited a more drastic effect on exon skipping in both *CD44v8* and *CD44v5* minigenes (Figs. 4d, e, Supplementary Fig. 4d). Conversely, siRNA-mediated silencing of hnRNPM showed a moderate but not significant increase of the AKAP8's splicing activity (Supplementary Fig. 4e, f). Taken together, these data indicate that AKAP8 abrogates hnRNPM's activity on exon skipping.

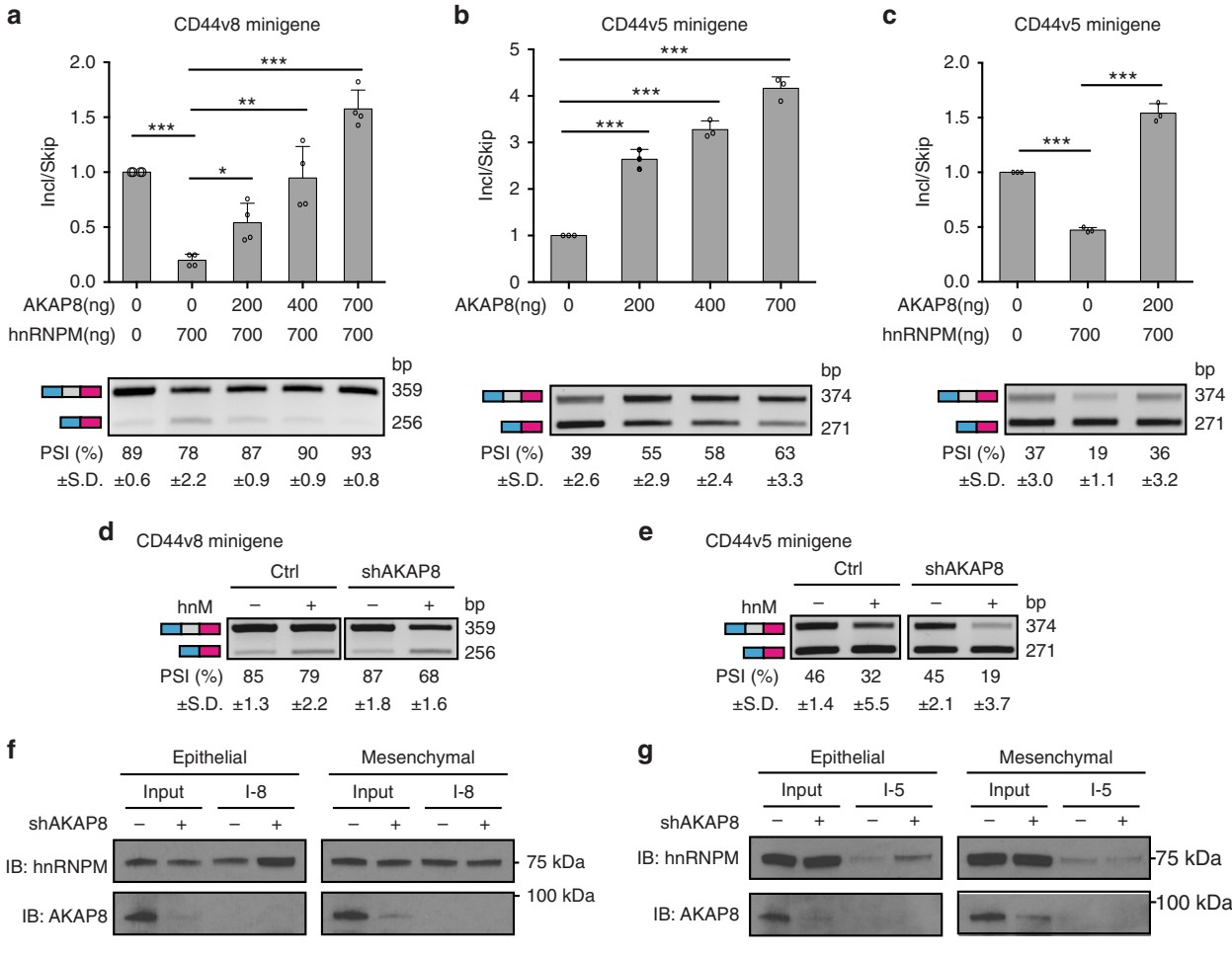

**Fig. 4 AKAP8 antagonizes hnRNPM's splicing activity. a** qRT-PCR (top) and semi-qPCR (bottom) analysis of CD44v8-splicing minigene assay showing that hnRNPM-mediated (hnM) exon skipping is antagonized by AKAP8 (AK8) in a dose-dependent manner. Incl: Inclusion. PSI: Percent Spliced In. **b–c** qRT-PCR (top) and semi-qPCR (bottom) analysis of CD44v5-splicing minigene assay showing that AKAP8 promotes v5 inclusion **b** and that AKAP8 inhibits hnRNPM-mediated exon skipping activity **c**. **d**, **e** Semi-qPCR of CD44v8 and CD44v5 minigene reporter assays showing increased exon skipping activity of hnRNPM in AKAP8 knockdown cells. **f**, **g** RNA pull-down assays showing that hnRNPM binds to its RNA cis-elements I-8 **f** and I-5 **g** more strongly in AKAP8-silenced epithelial HMLE cells. Twist-induced mesenchymal cells did not show binding differences. The sequences of I-8 and I-5 are shown at the bottom of each panel. Error bars indicate s.d. $N = 3$. (**) $P < 0.01$, (***) $P < 0.001$ between indicated groups. $P$ values were tested by Student's $t$ test, two-tailed, in **a**, **b**, **c**. Source data are provided as a Source Data file.

Previous work indicated that hnRNPM mediates *CD44* exon skipping via binding to consensus GU-rich sequences in introns downstream of exons v5 and v8[10]. Thus, we examined whether AKAP8 precludes hnRNPM from binding to its intronic consensus sequences by RNA pull-down assay (Fig. 4f, g). We used two sets of RNA oligos, I-5 and I-8, that contained hnRNPM-binding motifs in the introns downstream of the v5 and v8 exons, respectively. Interestingly, the binding signal of hnRNPM to both I-5 and I-8 RNA oligos was enhanced in AKAP8 knockdown epithelial cells (compare lane 4 to lane 3 in Figs. 4f, g, left panels), whereas remained the same in AKAP8 knockdown mesenchymal cells (Fig. 4f, g, right panels). These results suggest that AKAP8 diminishes the ability of hnRNPM to bind to its RNA targets in an epithelial cell state-specific manner. Surveying the expression level and localization of AKAP8 and hnRNPM in the context of EMT showed no obvious differences between epithelial and mesenchymal cells (Supplementary Fig. 4g, h), suggesting that other mechanisms may be involved in contributing to this epithelial cell state-specific activity. These possibilities could include post-translational modification of

AKAP8 and/or hnRNPM, changes in levels of other hnRNPM-binding partners identified in Fig. 1, and changes in expression of the previously reported splicing regulator ESRP1, which serves as an antagonist of hnRNPM's splicing activity[9,10].

**AKAP8 promotes epithelial state-associated alternative splicing.** AKAP8 was only recently reported as an RNA-binding protein capable of binding to and regulating alternative splicing[45]. Given our observation that AKAP8 interacts with hnRNPM and regulates *CD44* minigene splicing, we sought to interrogate whether AKAP8 regulates global alternative splicing, especially EMT-related alternative splicing. Thus, we performed deep RNA sequencing using the HMLE/Twist-ER cell lines that express control or AKAP8 shRNA in both epithelial and mesenchymal states. We identified AKAP8-regulated splicing alterations totaling 144 and 228 alternative splicing events in epithelial (Fig. 5a, top panel) and mesenchymal states (Fig. 5a, bottom panel), respectively (FDR < 0.05, |ΔPSI| ≥ 0.1, average junction reads per cassette event ≥ 20). Classification of the AKAP8-mediated alternative splicing showed that the majority of the events were

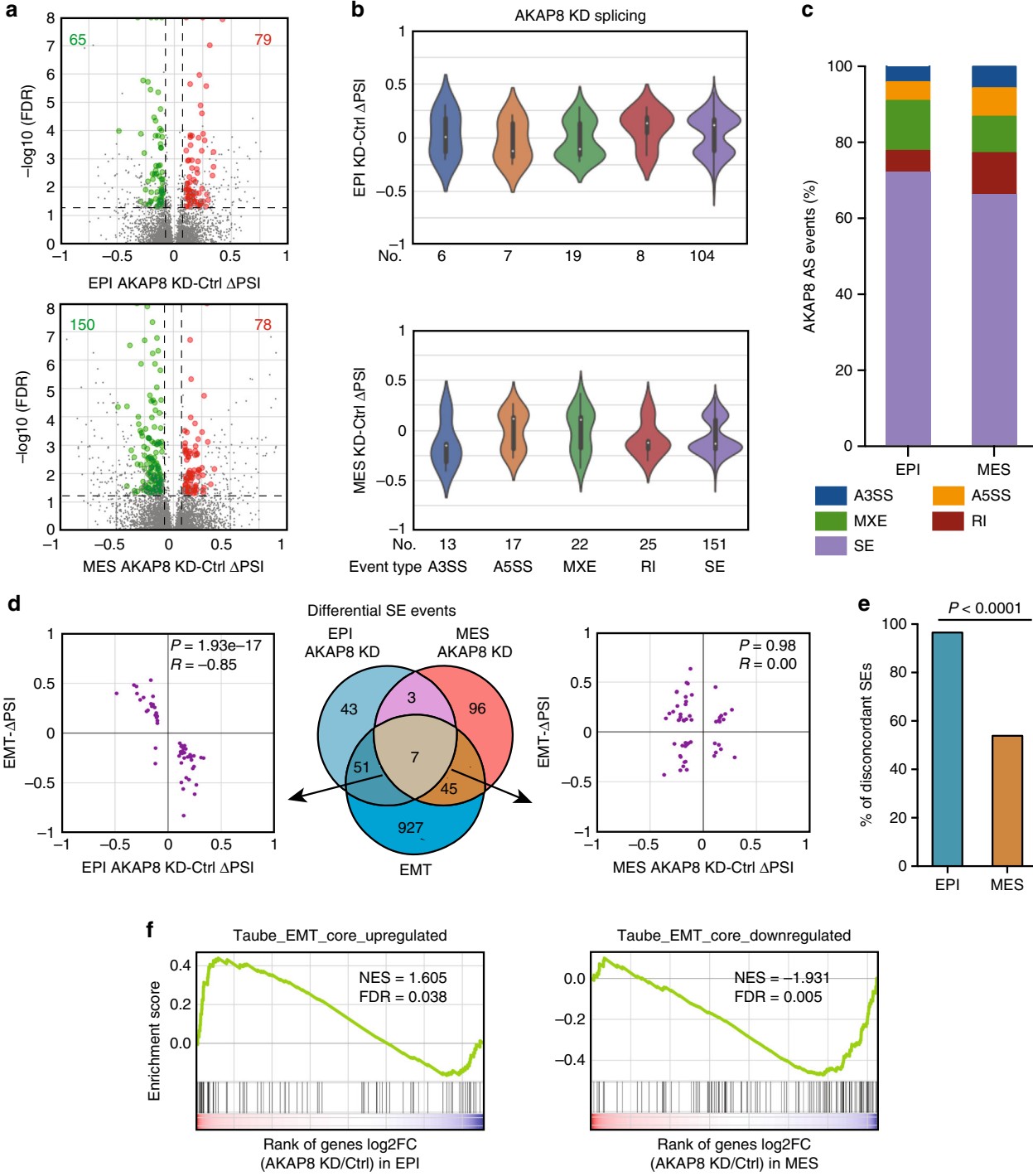

**Fig. 5 AKAP8 regulates EMT in a cell state-specific manner through alternative splicing. a** Volcano plots showing differential alternative splicing events identified upon knockdown of AKAP8 in HMLE/Twist-ER untreated epithelial cells (EPI, top panel) and in tamoxifen-induced HMLE/Twist-ER mesenchymal cells (MES, bottom panel). Significant alternative splicing events (FDR < 0.05 and |ΔPSI| > 0.1) were colored as green and red dots if AKAP8 knockdown resulted in skipping or inclusion, respectively. PSI: Percent Splice In. **b** Violin plot of differential alternative splicing distribution identified upon knockdown of AKAP8 in epithelial cells (top panel) and mesenchymal cells (bottom panel) grouped by alternative splicing types. A3SS: Alternative 3' Splice Site; A5SS: Alternative 5' Splice Site; MXE: Mutually Exclusive Exon; RI: Retention Intron; SE: Skipped Exon. **c** Stacked bar graph of the proportion of different types of AKAP8-dependent alternative splicing events in epithelial and mesenchymal cells. **d** AKAP8-dependent SE events overlap with SEs altered during EMT (middle panel). Scatterplots of ΔPSI values between EMT-regulated SEs and AKAP8-dependent differential SEs identified in epithelial cells (left panel) and mesenchymal cells (right panel). SEs regulated during EMT and by AKAP8 in epithelial cells are highly negatively correlated ($R = -0.85$, $P = 1.93e-17$), whereas those in mesenchymal cells are not correlated ($R = 0.00$, $P = 0.98$). P values were determined by hypergeometric test. **e** 96.6% (56/58) of epithelial AKAP8 differential SEs are discordant with EMT compared with 53.8% (28/52) of mesenchymal events. $P < 0.0001$ (P computed by Fisher's exact test). **f** AKAP8 knockdown in epithelial cells causes upregulation of genes that are upregulated during EMT (left panel), and AKAP8 knockdown in the mesenchymal state results in downregulation of genes that are downregulated during EMT (right panel). Source data are provided as a Source Data file.

cassette skipped exon (SE) events, the most common form of alternative splicing (Fig. 5b, c). Note that the gene expression levels of these AKAP8-mediated splicing events were not altered in response to AKAP8 knockdown in either the epithelial or mesenchymal state (Supplementary Fig. 5a, b).

As SEs represent the vast majority of AKAP8-regulated alternative splicing events, we next overlapped AKAP8-regulated SEs with those regulated during EMT. The latter were obtained from the differential splicing events in control HMLE/Twist-ER cells that are in either the epithelial state or the mesenchymal state. Our results revealed that more than half of AKAP8-regulated SEs in epithelial cells are also altered during EMT (58 overlapping events out of 104 AKAP8-regulated SEs). Remarkably, the vast majority (96.6%) of the 58 common events are regulated in a discordant direction, indicating that AKAP8 antagonizes EMT-associated alternative splicing (Fig. 5d, left panel). This inverse direction of regulation by AKAP8 is unique to the epithelial state. In mesenchymal cells, 52 out of the 151 AKAP8-regulated events overlapped with the SEs that are altered during EMT. However, they showed roughly equal concordant (46.2%) and discordant (53.8%) directions compared with EMT-associated SEs (Fig. 5d, right panel). The fraction of discordantly regulated events in the epithelial state is significantly higher than that in the mesenchymal state (Fig. 5e, $P < 0.0001$ by Fisher's exact test). These data show that AKAP8 strongly suppresses EMT-associated alternative splicing in a manner that is epithelial state-specific. This dichotomy is supported by the fact that only 10 SEs were found to be regulated by AKAP8 in both the epithelial and mesenchymal states (Supplementary Fig. 5c). These 10 overlapping events are all regulated in the same direction (Supplementary Fig. 5d). Interestingly, 7 out of the 10 events overlapped with the EMT-associated SEs and all showed discordant regulation with the EMT-associated SEs (Supplementary Fig. 5e). These results imply that AKAP8 antagonizes a set of EMT-associated SEs in both cell states. Experimental RT-PCR validation of SEs that are regulated by AKAP8 knockdown and during EMT showed that AKAP8 knockdown-mediated splicing alterations are consistent with the direction of splicing changes during EMT (Supplementary Fig. 5f). Further supporting the role of AKAP8 in inhibiting EMT came from Gene Set Enrichment Analysis (GSEA). Comparing the transcriptome of control and AKAP8 knockdown cells, we found that genes that are upregulated during EMT show increased expression upon AKAP8 knockdown in the epithelial state, whereas genes that are downregulated during EMT show decreased expression in the AKAP8-silenced mesenchymal cells (Fig. 5f). Together, these results show that AKAP8 antagonizes EMT-associated alternative splicing across the transcriptome to maintain an epithelial cell state.

**AKAP8 binds to RNA with a consensus motif**. To identify high-confidence binding sites for AKAP8 across the transcriptome, we performed single-nucleotide resolution enhanced cross-linking and immunoprecipitation (eCLIP). Two AKAP8 eCLIP biological replicates were performed in each of the epithelial and mesenchymal cell states and showed a high degree of correlation, highlighting the reproducibility of our assay (Supplementary Fig. 6a, b). By normalizing IP signal with size-matched input eCLIP libraries, we obtained quantitative estimates of AKAP8-binding intensity, resulting in 21,665 and 26,228 high-confidence AKAP8-binding sites in the epithelial and mesenchymal state, respectively (log2FC(IP/Input) $\geq 2$ per replicate, $-\log(\text{adjusted } P \text{ value}) \geq 3$, per replicate). Mapping the location of AKAP8-binding sites across the gene body revealed that the majority of binding sites are located in distal introns >500 nucleotides from

splice sites (Fig. 6a). Interestingly, although AKAP8 showed less binding to distal introns in the epithelial state compared with the mesenchymal state (Fig. 6b, $P = 9.38\mathrm{e}{-174}$ by Fisher's exact test), it binds to proximal intronic regions more significantly in the epithelial state (Fig. 6b, $P = 6.60\mathrm{e}{-116}$ by Fisher's exact test). These results suggest that AKAP8 regulates alternative splicing more directly in epithelial states through binding to pre-mRNA proximal intron regions. Differences in binding to other gene regions in the 5′ and 3′-UTR or the coding region showed no statistical differences (Fig. 6b).

Metagene analysis to assess AKAP8-binding intensity across all human introns and exons revealed significantly more frequent binding of AKAP8 to introns compared with exons (Fig. 6c). Interestingly, AKAP8 intronic binding in the epithelial state appears skewed towards the 5′ splice site while binding in the mesenchymal state is distributed more evenly across the intron (Fig. 6c, compare top and bottom plots in left panel). Quantification of the binding ability in the proximal introns revealed significantly higher binding in the epithelial state (Fig. 6d), reinforcing our observations that AKAP8 binds more frequently (Fig. 6b) and more strongly (Fig. 6c) to proximal intronic regions in the epithelial state.

Using the single-nucleotide precision of eCLIP, we identified AKAP8 high fidelity binding motifs. We took the center of each AKAP8-binding interval and extended the length 100 nucleotides upstream and downstream of each center. The AKAP8-binding motifs contain guanine stretches of at least three nucleotides flanked by one or two uridine or adenine nucleotides (Fig. 6e), and AKAP8 shares very similar recognition motifs in epithelial and mesenchymal states. We identified 16 and 20 SEs that are both bound by AKAP8 within 1000 bp of the variable exon and regulated by AKAP8 in epithelial and mesenchymal cell states, respectively (Supplementary Table 2). We did not observe a relationship between AKAP8-binding topology and direction of SE regulation.

Screening the AKAP8-binding peaks near the CD44v8 exon indicated one significant peak located upstream of the v8 exon containing two AKAP8 motifs with guanine stretches flanked by adenine (Supplementary Fig. 6c). AKAP8's binding was validated by RNA pull-down assay with an RNA oligo, AK, which was derived from the core of this peak (Supplementary Fig. 6d). The most enriched AKAP8-binding site is located at the upstream proximal intron of *CLSTN1* variable exon 11 containing four AKAP8 motifs. AKAP8 binds in both epithelial and mesenchymal states but with a greater than twofold binding intensity in the epithelial state (Fig. 6f). Because *CLSTN1* exon 11 inclusion increased during EMT and in AKAP8 knockdown epithelial cells, these results suggest that AKAP8 binds to *CLSTN1* and inhibits EMT-mediated *CLSTN1* exon inclusion, a regulatory axis which may block EMT.

**Depletion of the CLSTN1 short isoform promotes EMT**. To experimentally test the role of AKAP8-mediated *CLSTN1* splicing during EMT, we examined the binding of AKAP8 to the above eCLIP-predicted *CLSTN1*-binding motifs by RNA pull-down experiments. We found that AKAP8 binds to the consensus sequences in both 293FT cells and HMLE/Twist-ER cells (Fig. 7a, bottom panel), but not the mutated sequences (Fig. 7a).

CLSTN1 is a member of the calsyntenin family and was reported to function in cargo trafficking along neuronal axons. Thus far, the role of CLSTN1 or its isoforms in EMT has not been reported. We designed isoform-specific shRNAs to silence either the CLSTN1-L or CLSTN1-S individually (Fig. 7b, top panel) and achieved specific isoform knockdowns without showing decreased

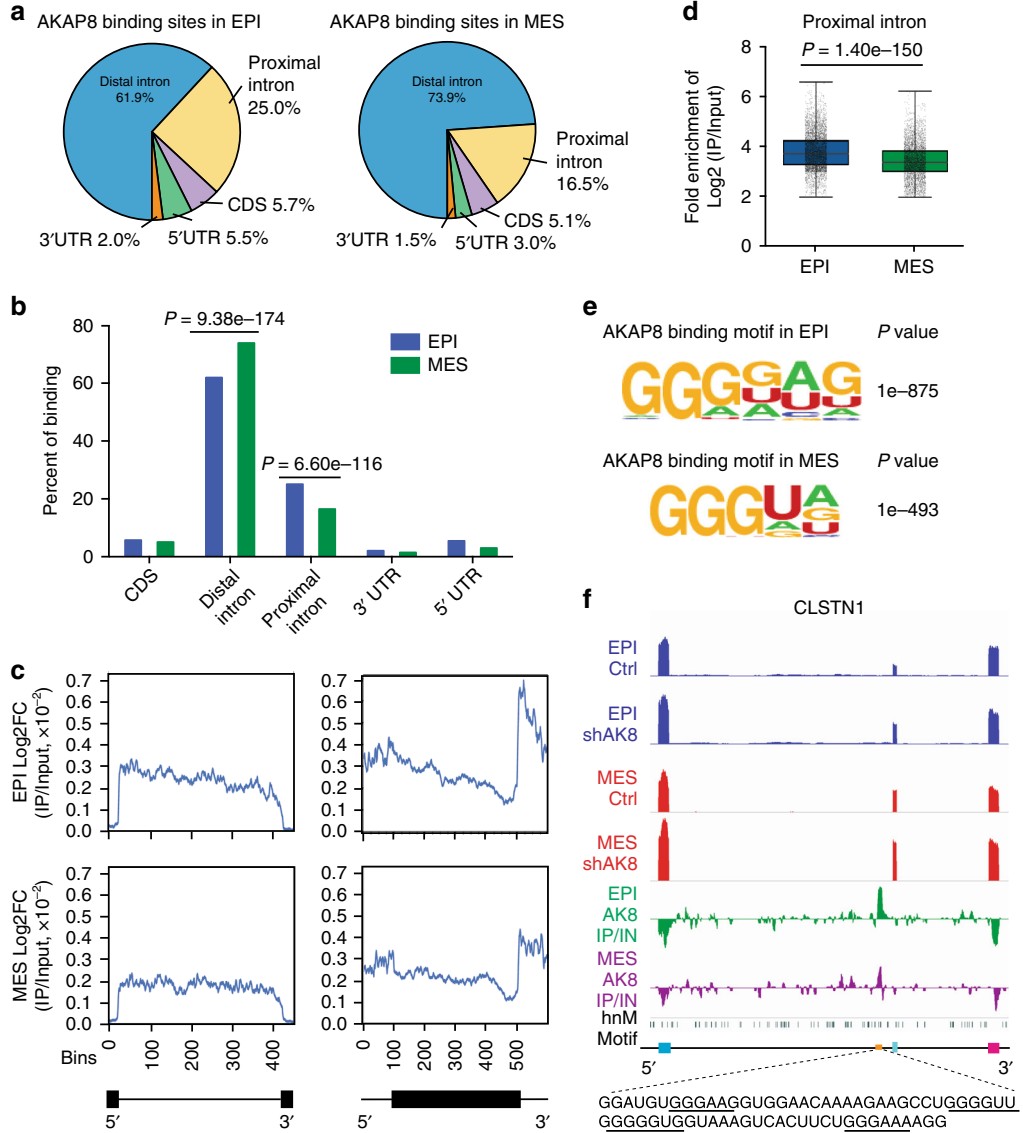

**Fig. 6 AKAP8 eCLIP identifies its RNA-binding targets and cis-element motifs. a** Distribution of AKAP8 eCLIP-binding sites across different regions of gene bodies. **b** Bar plot comparing the percentage of AKAP8-binding sites in each gene region between epithelial (EPI) and mesenchymal (MES) cell states. *P* values were computed by Fisher's Exact test. **c** Metagenes plotting AKAP8-binding intensity (average log2FC IP/Input) in epithelial state (top panels) or mesenchymal state (bottom panels) across all introns (left panels) or exons (right panels). Introns (black line flanked by two black rectangles) were scaled to 400 bins. The flanking 50 nucleotides were split into 25 bins. Exons (black rectangle flanked by two black lines) were scaled to 500 bins. The flanking 1000 nucleotides were split into 100 bins. **d** Boxplots with jitters comparing the binding intensity (log2FC IP/Input) of all AKAP8-binding sites located in proximal introns showing that AKAP8 binds more strongly to proximal introns in the epithelial state compared with the mesenchymal state (*p* computed by Student's *t* test, two-tailed). The line within each box represents the median. Upper and lower edges of each box represent 75th and 25th percentile, respectively. The whiskers extend across the entire range of the data. **e** Weblogos depicting the most significant AKAP8 eCLIP-binding motif in epithelial and mesenchymal cells identified through de novo motif analysis. The two motifs are similar with guanine stretches of at least three nucleotides flanked by uridine or adenine nucleotides. *P* values were determined by hypergeometric test. **f** Integrated genome viewer tracks centered on a cassette exon (exon 11) of CLSTN1, which undergoes exon inclusion upon AKAP8 (AK8) knockdown in the epithelial cells. Top four tracks represent autoscaled RPM-normalized RNA seq reads. Bottom two tracks represent IP/input normalized eCLIP signal. Black bricks indicate hnRNPM (hnM) binding motifs (GU-rich). Bottom cartoon shows CLSTN1 cassette exon in light blue flanked by upstream (dark blue) and downstream (pink) constitutive exons. Yellow bar indicates AKAP8 eCLIP-binding site. Zoomed out sequence indicates nucleotide sequence covered by binding site with AKAP8 motifs underlined. Source data are provided as a Source Data file.

expression of the non-targeted isoforms (Fig. 7b, bottom panel). TAM induction of EMT in the HMLE/Twist-ER control and CLSTN1-S shRNA-expressing cells showed that silencing CLSTN1-S-accelerated EMT significantly within 8 days (Fig. 7c). These cells also switched expression from epithelial markers to mesenchymal markers (Fig. 7d) and displayed loss of E-cadherin at the cell junctions (Fig. 7e). By contrast, control cells at this

stage were largely maintained in the epithelial state (Fig. 7c–e). These results reveal that silencing CLSTN1-S promotes EMT.

Although silencing CLSTN1-L did not show an overt difference compared with control cells, it caused significant cell death after the TAM induction for 4 days (Supplementary Fig. 7a), implying that CLSTN1-L is required for cell survival during EMT. As *CLSTN1* transcripts levels remain relatively consistent in

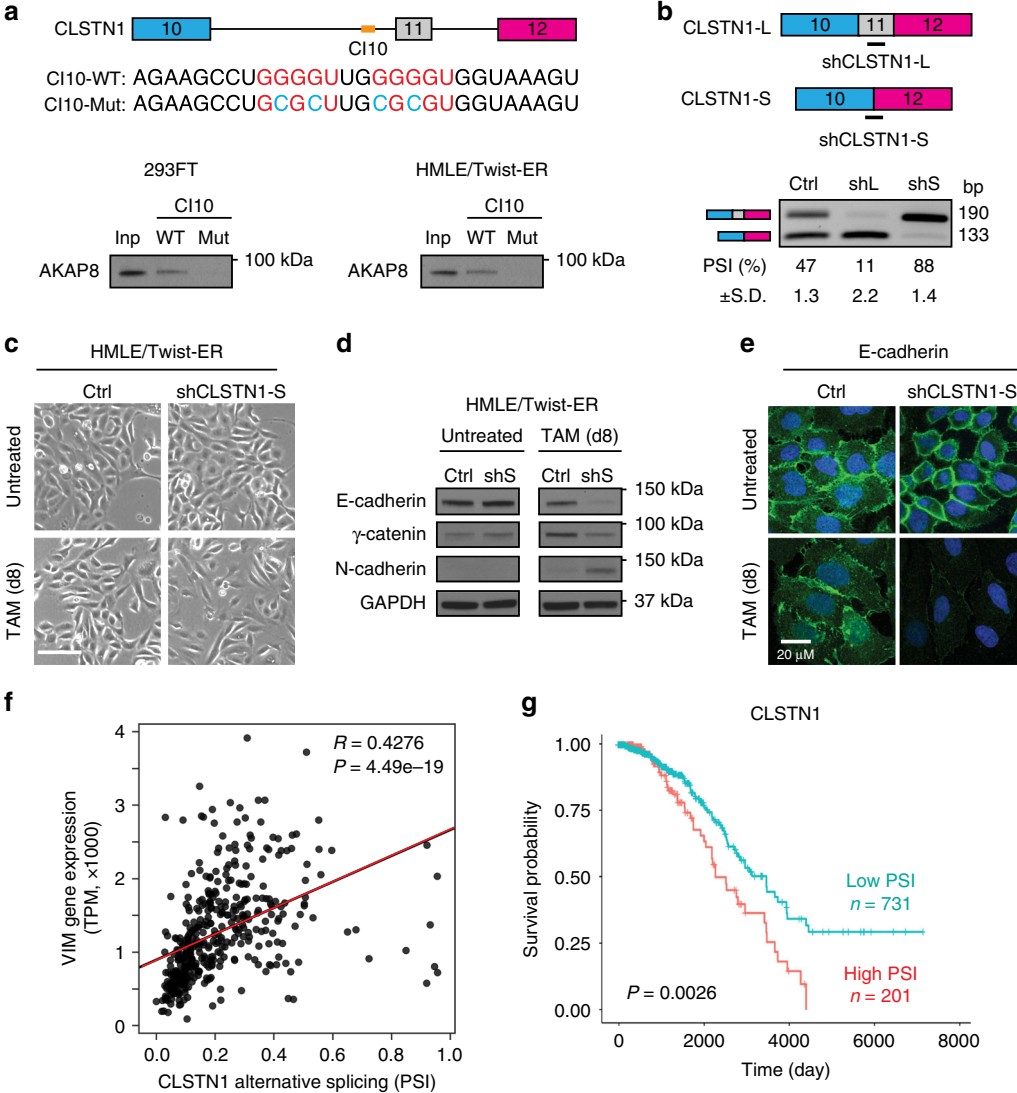

**Fig. 7 The CSLTN1 isoform with exon 11 inclusion promotes EMT. a** RNA pull-down assay showing that AKAP8 binds to the cis-element located in the intron upstream of CLSTN1 variable exon 11. A schematic of the CLSTN1 pre-mRNA and the cis-element sequence (CI10) are shown in the top panel. RNA pull down was performed using lysates from 293FT and HMLE/Twist-ER cells. Inp: Input. **b** A schematic of the location of CLSTN1 isoform-specific shRNAs (black line, top panel), and semi-qPCR analysis (bottom panel) showing CLSTN1 isoform-specific knockdown efficiency in HMLE/Twist-ER cells. shL: CLSTN1-L knockdown; shS: CLSTN1-S knockdown. **c** Phase-contrast images (× 10) indicating accelerated cell morphology changes in control shRNA (Ctrl) and CLSTN1-S knockdown (shCLSTN1-S) cells 8 days after tamoxifen (TAM) induction. White line represents scale bar at 100 μm. **d** Western blot analysis of EMT markers showing that CLSTN1-S silencing resulted in decreased expression of epithelial markers E-cadherin and γ-catenin and increased expression of the mesenchymal marker N-cadherin after 8 days of TAM induction. **e** Immunofluorescence images showing the loss of E-cadherin expression at cell junctions in the CLSTN1-S knockdown cells after 8 days of TAM treatment. **f** Plot of the mesenchymal marker vimentin (VIM) expression and CLSTN1 alternative splicing in the breast cancer TCGA data set showing a significant positive correlation in luminal breast cancer. PSI: Percent Spliced In. P value was determined by person correlation test. **g** Kaplan–Meier plot showing correlation between CLSTN1 alternative splicing, represented as PSI levels, and breast cancer patient survival in TCGA data set. The average PSI for the high group is 0.49 and low group is 0.15, which were determined by k-means clustering. P value was determined by log-rank test. Source data are provided as a Source Data file.

epithelial and mesenchymal cells (Supplementary Fig. 7b), these results indicate that isoform switching from CLSTN1-S to CLSTN1-L has an important role for cells to undergo EMT.

We then analyzed the levels of CLSTN1 isoforms using the TCGA breast cancer patient RNA-sequencing data, and found that higher fractions of CLSTN1-L, represented by high PSI values, were associated with higher levels of the mesenchymal marker vimentin in patient breast tumors, especially in luminal breast cancer (Fig. 7f and Supplementary Fig. 7c), These CLSTN1-L high patients also showed worse survival compared with those patients with lower levels of CLSTN1-L (Fig. 7g). Interestingly however, CLSTN1 expression did not correlate with

patient survival (Supplementary Fig. 7d). These results show that *CLSTN1* alternative splicing, rather than its gene expression, is significantly associated with breast cancer patient survival.

## Discussion

As an essential step in gene expression, alternative splicing contributes significantly to an ever-growing number of human diseases, especially to cancer[12,15]. In this study, we identified an RNA-binding protein AKAP8 as an alternative splicing modulator to inhibit cells from undergoing an EMT transition. AKAP8 was originally identified as a kinase anchoring protein

that recruits protein kinase A to nuclear matrix and chromatin structures[46–49]. It was later reported to function as a DNA-binding protein[50] as well as an RNA-binding protein that influences RNA stability and pre-mRNA splicing[45,51,52]. Our study provides a functional role for AKAP8 in RNA metabolism and connected it further to a tumor metastasis suppression phenotype. We show that AKAP8 is capable of inhibiting the splicing activity of hnRNPM, perturbing hnRNPM-mediated alternative splicing that occurs during EMT. AKAP8 is able to bind its own RNA consensus sequences and prevent the EMT-associated alternative splicing. Functionally, AKAP8 inhibits EMT and breast cancer metastasis to the lung. High levels of AKAP8 expression predicts a better survival of breast cancer patients. Thus, these results nominate AKAP8 as a splicing regulator for EMT-associated alternative splicing and for inhibition of EMT and tumor metastasis, highlighting the importance of RNA-binding protein in regulating cancer progression.

The identification of AKAP8 in suppressing EMT and cancer metastasis was through a biochemical approach to capture proteins that interact with hnRNPM, previously shown to promote EMT and tumor metastasis by regulating alternative splicing of CD44. AKAP8 interacts with hnRNPM and antagonizes hnRNPM's splicing activity on CD44 exon skipping. One interesting observation was that the AKAP8-hnRNPM interaction became stronger upon RNase treatment, which led us to speculate that AKAP8 binding to hnRNPM blocks hnRNPM from binding to its RNA targets. Supporting this view, depletion of AKAP8 promoted hnRNPM's ability to bind its consensus RNA sequences and to stimulate exon skipping. Previous characterization of AKAP8 showed that the N-terminal region of AKAP8 mediates its protein interactions with other splicing factors, such as hnRNPM[45]. The C-terminal domain of AKAP8 mediates pre-mRNA binding through two zinc finger domains. These results are congruent with our findings that AKAP8 interacts with hnRNPM, which in turn, inhibits hnRNPM-mediated splicing. AKAP8 is also capable of binding RNA and directly regulates alternative splicing. As both AKAP8 and hnRNPM share somewhat similar but not completely overlapping binding motifs, it is likely that they compete on binding to a subset of RNAs. Notably, the opposite splicing regulatory roles of AKAP8 and hnRNPM in co-regulation of EMT-associated splicing events provides increasing evidence that the complex interplay among RNA-binding proteins tightly controls alternative splicing[9].

Functionally, we used both a PDX model of tumor metastasis and a highly metastatic cell line LM2 and showed that AKAP8 knockdown promotes breast cancer lung metastasis and ectopic expression of AKAP8 inhibits metastasis. These results complement our previous findings showing that depletion of the CD44s splice isoform inhibits breast cancer metastasis. In this study, we have provided evidence on the role of AKAP8 in suppressing metastasis using immune-compromised mice. As tumor microenvironment plays important roles in both inhibiting and promoting tumor metastasis, future validation of these results in immune-competent mice will be needed to better understand the role of the RNA metabolism in metastasis.

Our transcriptome-wide RNA-sequencing analysis demonstrated that AKAP8 regulates alternative splicing with predominant regulation of SEs. Most of the AKAP8-regulated alternative splicing events in the epithelial state show opposite directions of splicing regulation compared with those occurring during EMT, suggesting a functional role of AKAP8 in antagonizing EMT-associated alternative splicing. In addition to AKAP8, there are a handful of RNA-binding proteins that have been functionally implicated during EMT, including hnRNPM, ESRP1, RBFOX2, QKI[9,10,29,32,53–56]. In the AKAP8 knockdown cells, we did not identify any drastic changes in expression of

these RNA-binding proteins, suggesting again that AKAP8 has a direct role in regulating alternative splicing, resulting in inhibition of EMT.

The eCLIP analysis of AKAP8 binding revealed the binding sites of AKAP8 across the transcriptome. AKAP8 preferentially binds to introns and less frequently binds to the coding regions or 5′- or 3′-UTRs. Our eCLIP results resolved a short 5–6-mer consensus motif consisting of guanine stretches of at least three nucleotides long preferentially flanked by adenine or uridine nucleotides. The top motifs were many orders of magnitude more significant than secondary motifs identified by the de novo motif analysis pipeline and were consistent in both epithelial and mesenchymal cells, suggesting that AKAP8 binds a similar motif regardless of cell state and that our motif analysis method is reproducible. A previous study[45] reported an AKAP8 motif through de novo motif analysis, where three relatively disparate sequences of 12-mer motifs were obtained. Although these motifs are longer than the ones we report, our findings are somewhat consistent with an AGGAGGA sequence identified in one of the motifs in that study. This commonality notwithstanding, we speculate that the motifs derived from our study resolved a more precise AKAP8 motif with a higher level of statistical significance provided by the single-nucleotide resolution of the eCLIP methodology and the integration of input normalization into our AKAP8-binding site calling compared with the less-precise RIP-seq method used in the aforementioned study. Furthermore, we resolved the motifs using a background control of shuffled human introns, the gene region AKAP8 binds the most abundantly.

We have experimentally characterized one of the newly identified AKAP8 splicing targets, CLSTN1. CLSTN1 is a transmembrane protein and belongs to the calsyntenin family, a subset of the cadherin superfamily[57]. CLSTN1 was found mainly participating in neural axon trafficking and branching as well as acting as a mediator of virus replication[58–60]. This study made an unexpected functional connection between CLSTN1 alternative splicing and EMT. We showed that silencing of the CLSTN1-S isoform, a product of AKAP8-regulated splicing, accelerates EMT, suggesting that the CLSTN1-S splice isoform is critical for maintaining an epithelial state. By contrast, knockdown of the CLSTN1-L isoform resulted in cell death, suggesting a necessity of this isoform for cells to undergo EMT. Furthermore, as the CLSTN1 splice isoform levels, but not its transcription level, correlates with patient survival, we speculate that tipping the AKAP8 downstream target from CLSTN1-L to CLSTN1-S may be an effective strategy for the treatment of breast cancer. Future studies on the role of CLSTN1 splice isoforms in breast cancer metastasis will be necessary to determine whether this splice isoform switch is functionally important for metastasis, as we previously uncovered for the CD44 splice isoform switch[8,26].

In conclusion, this work has led to the identification of a new role of the RNA-binding protein AKAP8 in suppressing EMT and breast cancer metastasis. We demonstrated that the CLSTN1-S splice isoform, generated by AKAP8-mediated alternative splicing, inhibits EMT and shows an inverse correlation with breast cancer progression. Our results reveal the complex regulation of alternative splicing by RNA-binding proteins in cancer and suggest the power of studying alternative splicing to uncover a new layer that regulates tumor metastasis.

## Methods

**Cell cultures and EMT induction.** Human embryonic kidney cell line 293FT (ATCC), colorectal carcinoma cell line HCT116 (ATCC), breast carcinoma cell line MDA-MB-231 derivative LM2 (from Dr. Yibin Kang at Princeton University), and PDX-derived HIM3 (provided by Dr. Helen Piwnica-Worms at MD Anderson) cell lines were cultured in DMEM supplemented with 10% FBS, L-glutamine, penicillin, and streptomycin. HMLE/Twist-ER cells (from Dr. Jing Yang at UCSD) were grown in Mammary Epithelial Cell Growth Medium (Lonza, USA). To induce

EMT in HMLE/Twist-ER cells, a final concentration of 20 nM TAM was added to its culture medium, and cells were split every other day until the mesenchymal morphology was fully observed.

**BioID pull down assay and mass spectrometry analysis**. A full-length hnRNPM cDNA was cloned into pQCXIP-BirA(R118G)-HA at Not1 and BamH1 site to express an hnRNPM-BirA-HA fusion protein. An hnRNPM-BirA-HA expressing stable cell line was generated in 293FT cells. BioID pull-down experiments were performed according to a previously published method[61]. To describe, cells in a 15 cm dish were pretreated with 50 μM biotin 24 hours prior to the collection. Cells were washed three times with phosphate-buffered saline (PBS) and scraped directly in 900 μl lysis buffer (50 mM Tris-HCl, 500 mM NaCl, 0.2% PBS, 1 mM dithiothreitol (DTT), fresh added protease inhibitors) and a final concentration of 2% Triton X-100 was added, followed by sonication. After diluting the lysates with an equal amount of pre-chilled 50 mM Tris-HCl pH 7.4, a second round of sonication was performed. The lysates were centrifuged and the supernatant was incubated with 150 μl strep-tavidin beads at 4 °C overnight by rotation. After incubation, the beads were pelleted by centrifugation and washed twice in buffer 1 (2% SDS); twice in buffer 2 (0.1% deoxycholic acid, 1% Triton X-100, 1 mM EDTA, 500 mM NaCl, 50 mM Hepes pH 7.5); twice in buffer 3 (0.5% deoxycholic acid, 0.5% NP-40, 1 mM EDTA, 250 mM LiCl, 10 mM Tris-HCl pH 7.4); twice in buffer 4 (2 M urea, 10 mM Tris-HCl pH 8.0). The proteins were eluted in 90 μl elution buffer (2 × SDS sample buffer supplemented with 20 mM DTT and 2 mM biotin) by boiling for 10 min. 10% of the samples were analyzed by silver stain and 90% were subjected to mass spectrometry analysis at the Harvard Taplin Mass Spectrometry Facility.

**Plasmids and shRNAs**. Twenty-nine ORFs of splicing factors were obtained as ORF entry clones. They were then cloned into a pLenti6.3 V5/Dest vector to generate the destination clones by the gateway LR reaction (Invitrogen, USA). The 29 splicing factors were selected based on the rank of the unique peptide reads of Mass Spec and the availability of the clones in our cDNA library[62,63]. All plasmid constructs were validated by DNA sequencing. The AKAP8 cDNA were cloned into PCDH-CMV-MCS-EF1-Puro between XbaI and BamHI sites with a Flag-tag fused at the C-terminal and were used in the *CD44v8* splicing minigene assay. The AKAP8 cDNA was also cloned into a DOX-inducible plasmid pCW57-GFP-2A-MCS between MluI and BamHI sites and were used for over-expression in LM2 cells in the in vivo metastasis assay. For AKAP8 reexpression in the AKAP8 KD HCT116 cells, three synonymous mutations were introduced to AKAP8 shRNA-1 targeting site (wt: GCCAAGATCAACCAGCGTTTG, mut: GCCAAGATTAATCAACGTTTG), by Q5 Site-Directed Mutagenesis kit (NEB, E0554S). All shRNAs were design using the ranidesigner program at Life Technologies and isoform-specific shRNAs were designed according to the described principles[64]. ShRNAs were cloned into pLKO.1 vector. All shRNA sequences were listed in Supplementary Table 1.

**Immunoblotting assay**. Cells growing in tissue culture dishes were washed twice with cold PBS and collected in radioimmunoprecipitation assay buffer (20 mM Tris-HCl, pH 7.4, 150 mM NaCl, 1% Triton X-100, 0.5% sodium deoxycholate, 1% SDS, 1 mM EDTA, 1 mM NaF, 1 mM Na3VO4, 1 × protease inhibitor cocktail). After incubating on ice for 15 min, lysates were clarified by centrifuge at 12,000 rpm for 10 min, 4 °C. Protein concentrations were quantified by bradford protein assay (Bio-Rad, Catalog no. 500-0006) and boiled in final concentration of 1 × sodium dodecyl sulfate sample buffer. Equal amounts of protein were subjected to electrophoresis and transferred to methanol activated PVDF membrane (Millipore, #IPVH00010). Membranes were blocked in 5% non-fat milk in TBST (Tris-HCl buffer, pH 7.4, supplemented with 0.1% Tween-20) for 1 hour at room temperature and followed by primary antibody incubation overnight at 4 °C. Antibodies used in this study and their dilutions were listed at respective dilutions: Flag (Sigma, F1804, 1:2000), RBM10 (One World Lab, 1:1000), hnRNPR (One World Lab, 1:1000), RBMX (Cell signaling, 14794, 1:1000), hnRNPF (Santa Cruz, sc-390048, 1:200), AKAP8 (Abcam, ab72196, 1:500), PTPB1 (One World Lab, 1:1000), E-cadherin (Cell Signaling, 3195, 1:1000), γ-catenin (Cell Signaling, 2309, 1:1000), Occludin (Abcam, ab168986, 1:500), FN1 (BD, 610077, 1:2000), N-cadherin (BD, 610920, 1:1000), hnRNPM (Origene technologies, TA301557, 1: 50,000), GAPDH (EMD Millipore, MAB374, 1:10000), β-actin (Sigma, A5441, 1:10000). Then, membranes were washed with TBST for three times, 5 min each and incubated at corresponding HRP-conjugated secondary antibodies for 1 hour at room temperature. After four times washing with TBST, the targeted bands were developed with ECL (ThermoFisher, PI32106) and detected either with film or ChemiDoc Imaging Systems. All uncropped scans were provided in the Source Data file.

**CD44 splicing minigene assay**. *CD44* splicing minigene assay were carried out in 293FT cells. In all, $2.25 × 10^5$ cells were plated in each well of a 24-well plate 24 hours prior to transfection, and were co-transfected with RNA-binding protein plasmids and *CD44v8* or CD44v5-splicing minigene reporter using Lipofectamine 2000 (Invitrogen). Cells were collected 24 hours after transfection for RNA extraction (Omega Bio-Tek) and reverse transcription, followed by qRT-PCR or semi-qPCR to examine spliced isoforms. For qRT-PCR analysis of splicing, isoform-specific primers were used to detect the inclusion and skipping isoforms,

respectively. The ratio of the inclusion to skipping was calculated and compared. Because PCR from two primer sets can give rise to different amplification efficiency, we avoid taking the sum of the values of two PCR products as a denominator for calculating the percent spliced in (PSI). For semi-qPCR experiments, a same set of primers was used to amplify both the inclusion and skipping products, which were resolved by agarose gel electrophoresis.

**Immunoprecipitation assay**. 293FT cells overexpressing Flag-hnRNPM-HA were lysed in CoIP lysis buffer (20 mM Tris-HCl pH 7.5, 137 mM NaCl, 10% glycerol, 1% NP-40, 2 mM EDTA, 20 mM NaF, 1 mM NaVO3, and 20 mM β-glycerophosphate with fresh PI added). After centrifugation, lysate protein concentration was quantified. The lysate was pre-cleared with sepharose beads and incubated with corresponding antibodies at 4 °C overnight with rotation. Next day, protein A beads were added to capture the antibody–protein complexes and rotated at 4 °C for 3 hours. Beads were washed with lysis buffer for three times, followed by protein elution in 2 × sodium dodecyl sulfate sample buffer and boiled for 10 min. Samples were analyzed by western blot for testing IP efficiency and interacting proteins with corresponding antibodies.

**RNA pull-down assay**. RNA oligonucleotides labeled with biotin at the 5'-end were synthesized by Integrated DNA Technologies. The RNA sequences used in this study were listed in the following, I-8: GCUUUGGUGGUGGAAUGGUGCUAUGUGG, I-5: UGGCGGUCGGCAGUUCUGGGGUUAGAUGA, AK: GGUUGGUAAGGGGG AGG GGAUAAAAUGGUG, NC: GCUUUGAUGAUGAAAUGA, CI10-WT: AGA GCCU GGGGUUGGGGGGUGGUAAAGU, CI10-MUT: AGAAGCCUGCGCUUG CGCGUGG UAAAGU. In all, 400 pmol Biotinylated RNA oligos were conjugated with 50 μl of streptavidin beads (50% slurry; ThermoFisher) in a total volume of 300 μl of RNA-binding buffer (20 mM Tris, 200 mM NaCl, 6 mM EDTA, 5 mM sodium fluoride and 5 mM β-glycerophosphate, PH 7.5) at 4 °C in a rotating shaker for 2 hours. After three times wash with RNA-binding buffer, RNA-beads conjugates were incubated with 100 μg of nuclear extracts in 500 μl RNA-binding buffer at 4 °C in a rotating shaker overnight. Beads were then washed with RNA-binding buffer for three times and the RNA pull-down samples were eluted with 2 × SDS loading buffer for western blot analysis.

**Immunofluorescence assay**. Cells plated on coverslips were fixed with 4% paraformaldehyde in PBS for 10 min at room temperature, followed by permeabilization with 0.2% Triton X-100 for 5 min. Cells were blocked with 1% BSA in PBS and incubated with primary antibody overnight at 4 °C (1:100 dilution for E-cadherin, 1:100 for AKAP8, and 1:500 for hnRNPM). After three times wash with PBS, secondary goat anti-rabbit AlexaFluor 488 or goat anti-mouse AlexaFluor 568 (ThermoFisher, 1:500 dilution) were added for 1 h incubation at room temperature. After four times wash with PBS, coverslips were mounted with ProLong Gold anti-fade (ThermoFisher Scientific). Images were captured on Zeiss LSM 880 Confocal Microscope for E-cadherin using × 40 oil objective and Echo Revolve Microscope for AKAP8 and hnRNPM using × 20 objective.

**RNA-sequencing and data analysis**. Three biological replicates for control HMLE/Twist-ER with and without TAM treatment, three biological replicates for AKAP8 knockdown HMLE/Twist-ER without TAM treatment, and two biological replicates for AKAP8 knockdown HMLE/Twist-ER with TAM treatment cells were collected in 1 ml TRIzol for a 10 cm dish. RNAs were extracted followed by the TRIzol Reagent kit from Invitrogen. The purified RNAs were submitted to Genomic Facility at University of Chicago for RNA quality validation, poly(A) selected RNA-seq library generation and paired-end sequencing on a HiSeq 4000. RNA-seq reads were aligned to the human genome (GRCh37, primary assembly) and transcriptome (Gencode version 24 backmap 37 comprehensive gene annotation) using STAR version 2.6.1a[65] with the following non-standard parameters–outFilterMultimapNmax 1–outSAMstrandField intronMotif–outFilterType BySJout–alignSJoverhangMin 8–alignSJDBoverhangMin 3–alignEndsType EndToEnd. Only uniquely aligned reads were retained for downstream analysis.

Differential alternative splicing was quantified using rMATS version 4.0.2[66] with the following non-default parameters–readLength 100–cstat 0.01–libType fr-secondstrand. To identify significant differential splicing events, we set up the following cutoffs: FDR < 0.05, ΔPSI ≥ 0.1, and average junction reads per event per replicate ≥ 20. Differential gene expression analysis was performed by counting reads over genes from the same annotation as alignment using featureCounts version 1.5.0 with the following non-default parameters -s 2 -p -C -B. Differential gene expression analysis was conducted using DESeq2 performed on genes with read abundance larger than 10 counts over the smallest library size of all samples analyzed[67]. Significantly regulated genes were defined as genes with an | log2FC | > 1 and FDR < 0.05. GSEA was conducted using the GSEA pre-rank method where differentially expressed genes were ranked by log2FC before conducting GSEA analysis using gene set level permutation 10000 times[68,69].

**eCLIP assay and data analysis**. AKAP8 single-end eCLIP was performed in HMLE/Twist-ER cells without or with TAM treatment. Experiments were done in two biological replicates, following the protocol previously published[70]. Specifically,

input and AKAP8 antibody IP-ed fractions were run on an SDS-PAGE. Protein-RNA complexes between 95 kDa and 180 kDa were collected for RNA isolation followed by library generation. eCLIP libraries were sent to Genomic Facility at University of Chicago for single-end sequencing. eCLIP data processing was conducted using the public eCLIP pipeline version 0.2.1a (https://github.com/YeoLab/eclip/releases/tag/0.2.1a) and public merge-peaks pipeline version 0.0.6 (https://github.com/YeoLab/merge_peaks/ releases/tag/0.0.6), derived from the previously published eCLIP pipeline[70]. High-confidence eCLIP peaks for each cell state were called by selecting AKAP8-binding peaks with a minimum log2FC IP/Input signal > 2 among the replicates and an adjusted $p$ value < 0.001, resulting in 21,668 and 26,230 AKAP8-binding peaks in the epithelial and mesenchymal states, respectively. These were the peaks used for downstream analysis. De novo motif analysis was conducted by extending 100 nt on either side of the center of the peaks using HOMER v4.10 findMotifsGenome.pl script with the following non-default parameters -p 4 -rna -S 10 -len 4,5,6 -size 100 -chopify. De novo motifs were computed compared to a background of shuffled human introns. Metagenes and other analyses were computed using custom R and python scripts.

**In vivo metastasis assay**. All animal procedures were performed with approval from the Institutional Animal Care and Use Committee at Baylor College of Medicine. In total, $1 \times 10^5$ HIM3 control and AKAP8 knockdown cells or $2 \times 10^5$ LM2 control and AKAP8 OE cells were injected into 6–8 weeks old NSG nude mice by tail vein. Cell amount injected was quantified by D-luciferin injection and IVIS spectrum imaging (Caliper LifeScience), immediately after the injection as a reference signal. For the LM2 cells with tet-ON AKAP8 overexpression, mice were fed with DOX water (2 mg ml$^{-1}$) at all time through the experiment. The lung metastasis BLI signals were monitored every week till the tumor burdens were intolerable. Mice were euthanized and lungs were collected by 4% paraformaldehyde perfusion, followed by fixation and H&E staining to analyze the metastasis nodules. The areas of lung metastases were quantified by image J.

**Cell proliferation and wound healing assay**. For cell proliferation assay, 15,000 HIM3 control and AKAP8 knockdown cells were plated into a well of 96 well plates. Six hours later, cells were attached to the bottom, and the plates were loaded into the IncuCyte Zoom Live-content imaging system (Essen Bioscience). The cell confluences were scanned every 8 hours for a 4-day duration. Cell confluences were calculated for proliferation curve analysis.

For wound healing assay, $1 \times 10^6$ LM2 control or AKAP8 OE cells were seeded in each well of a six-well plate. In all, 24 hours later, a scratch was created in each well with a 200 μl tip and floated cells were washed way with PBS, and medium was refreshed into Dulbecco's Modified Eagle Medium (DMEM) without serum. The plates were placed into the IncuCyte Zoom Live-content imaging system for scanning at a 4 hour interval for 24 hours to collect scratch images. Percentage of wound healed was quantified by Image J.

**Patient data analysis**. AKAP8 expression levels downloaded from METABRIC database were Z score transformed and compared in different breast cancer subtypes, including Luminal A (LumA), Luminal B (LumB), HER2, Claudin low (CLOW), Basal, ER positive (ER+) and ER negative (ER−). AKAP8 expression correlation with breast cancer patient overall survival was separated by AKAP8 mean expression ($n = 1758$) from the METABRIC data set. The correlation between AKAP8 expression level and the distal metastasis-free survival was calculated using microarray data from a breast cancer cohort (GSE20685, $n = 327$), by setting online Kaplan–Meier plotter tool for optimal cutoff for separation of patients into high- and low-gene expression groups[38,71].

To examine the correlation between CLSTN1 splice isoform expression and breast cancer patient survival outcome, CLSTN1 exon 11 PSI values in patient samples were calculated according to previous analysis[9]. Kaplan–Meier survival analysis was conducted between the high PSI and low PSI groups, defined by $k$-means clustering ($k = 2$), using overall survival. $P$ values were computed using log-rank tests. CLSTN1 gene expression correlation with patient survival was calculated within the same patient cohort using $k$-means clustering ($k = 2$).

**Statistical analyses**. All data were presented as mean ± standard deviation, unless specifically indicated. Statistical significance was determined by two-tailed Student's $t$ test, unless specifically indicated. $P$ value < 0.05 was considered statistically significant. $P < 0.05$ (*), $P < 0.01$ (**), $P < 0.001$ (***) were indicated.

**Reporting summary**. Further information on research design is available in the Nature Research Reporting Summary linked to this article.

## Data availability

The RNA-sequencing and eCLIP data have been deposited in the Gene Expression Omnibus database under the accession code GSE139074 (https://www.ncbi.nlm.nih.gov/geo/query/acc.cgi?acc = GSE139074). The METABRIC data referenced during the study are available in a public repository from the cBioPortal website (http://www.cbioportal.org/). The distal metastasis-free survival of a breast cancer cohort (GSE20685) was downloaded from the Kaplan–Meier plotter website (https://kmplot.com/ analysis/index.

php?p = service&cancer = breast). The TCGA gene expression data set is available at (https://www.ncbi.nlm.nih.gov/geo/query/acc.cgi?acc = GSE62944). The TCGA alternative splicing data set is available at (https://gdc.cancer.gov/about-data/publications/PanCanAtlas-Splicing-2018). The source data underlying Figs. 1b–c, 2a, e, 3a, c, e, g, j, 4a–g, 5e, 6d, 7a, b, d and Supplementary Figs. 2a, c, 3a–c, e, 4a–h, 5e, 6a–d, 7b are provided as a Source Data file. All other data supporting the findings of this study are available within the article and its supplementary information files and from the corresponding author upon reasonable request.

## Code availability

All code used in this study is available upon request.

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

## Acknowledgements

We thank Dr. Eric L. Van Nostrand and Dr. Gene W. Yeo for their extensive help with the eCLIP assay and analysis. We thank Dr. Vladimir I Gelfand for providing the pQCXIP-BirA(R118G)-HA plasmid and the BioID protocol. This research was supported in part by grants from the US National Institutes of Health 5F30CA196118 (to S. E.H.), R01 CA182467, R01GM110146, R35GM131876 (to C.C.). C.C. is a CPRIT Scholar in Cancer Research (RR160009).

## Author contributions

X.H. and C.C. designed the study. X.H. initiated the project and performed all the experiments. J.L. helped with the revision experiments. X.H. organized all figures with help from S.E.H. on primary bioinformatics figures. X.H. and C.C. wrote the manuscript with edits from S.E.H. and inputs from co-authors. S.E.H. and R.Z. performed bioinformatics analysis. C.L.G. and K.L.S. provided cDNA plasmids and helped with cloning strategies for RBP screening. E.P. and H.P.W. provided reagents and protocol for the PDX xenograft model. C.C. conceived the project.

## Competing interests

The authors declare no competing interests.
