## [Peer Review File · Nature Communications]

Reviewers' Comments:

Reviewer #1:

Remarks to the Author:

Hu et al. report the identification of AKAP8 as an RNA binding protein that inhibits EMT and breast cancer metastasis through the regulation of alternative splicing. They demonstrate that AKAP8 interacts with hnRNPM and compete with hnRNPM to stimulate exon skipping of CD44. They show that AKAP8 is capable of directly binding to RNA and modulates alternative splicing events. They assess the function of one an AKAP8 splicing target, CLSTN1, in EMT.

Findings from this study contribute to improving our understanding of the regulators of breast cancer metastasis and highlight the role of splicing regulators in EMT and tumor progression. The data presented here is very clear and provides novel insights into the role of AKAP8 in metastasis.

A few questions should be addressed before the manuscript is ready for publication:

MAJOR COMMENTS:

- 1) Fig.1- Is there a difference in survival in breast tumors with high- vs. low-AKAP8 expression if subdivided by tumor subtype? Additionally, is there a correlation between AKAP8 and HNRMP expression in breast tumors?
- 2) Fig. 4- The authors nicely demonstrate that increased levels of AKAP8 can affect CD44 splicing in presence of HNRNPM, and reciprocally that decreased AKAP8 levels can have the opposite effect. Given that AKAP8 binds to HRNPM, that AKAP8 also binds to RNA, it remains unclear if AKAP8 can also modulate CD44 splicing in cells lacking HNRNPM, thus demonstrating that it can bind directly, or whether AKAP8 affects splicing by sequestering HNRNPM away from that transcript? This should at least be discussed.
- 3) Fig 5. and Fig. 6- The authors demonstrate that AKAP8 affects splicing of a set of target isoforms, and that AKAP8 binds directly to an RNA motif. How many of the spliced isoforms regulated by AKAP8 as detected by RNA-seq (Fig. 5) are also bound by AKAP8 as detected by eCLIP (Fig. 6)? Is there are difference in the position of AKAP8 binding site in exon that are included vs. skipped? Additionally, the authors should provide the location of AKAP8 and HNRNPM binding sites in the CD44 and CLSTN1 transcripts and discuss whether these two proteins are likely to compete for binding to the same sites?
- 4) The authors demonstrate that AKAP8 and HNRPM both regulate CD44 isoforms using a splicing reporter minigene. Using RNA-seq data from cells with AKAP8 or HNRPM KD have the authors been able to identify other targets that are also co-regulated by both?
- 5) The author previously uncovered a co-regulation of HNRPM and ESRP1 splicing targets (Harvey et al. RNA 2018). This begs the question whether ESRP1 or other RBP levels are affected by AKAP8 KD or OE. This should at least be discussed.
- 6) Fig. 4f, g – The author state that “hnRNPM’s binding signal was not altered in AKAP8 knockdown cells in the mesenchymal state [...] - suggested that post-translational modification of AKAP8 and/or hnRNPM are involved in contributing to this epithelial cell state-specific activity, an area of research that requires further exploration.”. Here again, the authors should at least discuss the possibility that these differences are mediated by changes in levels of other hnRNPM binding partners identified in Fig. 1 or levels of other co-regulators, such as ESRP1 identified in Harvey et al. 2018.

MINOR COMMENTS:

- 1) Please italicize gene and transcript names, e.g., CD44 gene or minigene throughout the manuscript.
- 2) Fig. 1- The experiments presented in Fig. 1 and associated supplemental data should be described more clearly. For example, what exactly is the assay used for screening candidate splicing factors? Is the assay's read-out fluorescence or CD44 isoform levels as detected by RT-PCR? How were the 29 SFs selected from the 51 candidates identified by BioID? What are the expression levels of the 29 SFs?
- 3) Fig. 1d and Fig. 7g- Please include number of patients in each subclass for survival plots.
- 4) Fig. 3 - Please provide data from the ctrl plasmid +/- DOX to control for dox-induced changes in Fig. 3g. Please also provide quantification and number of samples for histology analysis shown in Fig. 3j.
- 5) Fig. 4 -The difference in splicing ratio detected by QPCR vs. semi-quantitative PCR are following the same trend, but the values are quite difficult to interpret. Please clarify, or normalize the QPCR value to total transcript expression and plot a PSI value between 0-100%.
- 6) Supplementary Fig. 5a- The figure does not really answer the question how many of the changes in spliced isoform are also associated with changes in gene expression levels. Could the author provide a Venn diagram with overlapping gene IDs from the differentially spliced isoform and differentially expressed gene lists?
- 7) Fig. 5d,e - Where are the EMT AS events derived from ? Please add description and reference to dataset.
- 8) Fig. 6c- Please scale panels to the same values to allow comparison.
- 9) Fig. 6f- Please show IP signal as fold enrichment over input
- 10) Fig. 7f- Here the authors use vimentin levels as a mesenchymal marker. What is vimentin expression in cell lines used throughout the paper? Why not use E-cadherin as all other figures?
- 11) Fig. 7g- Please indicate the cut-off used for defining tumors as low vs. high CLSTN1 PSI.

Reviewer #2:

Remarks to the Author:

This is a well written and articulated article that documents observations that the A-Kinase Anchoring protein AKAP8/AKAP95 participates as a novel metastasis suppressor by antagonizing EMT responsive alternative splicing of RNA. Similar roles for this scaffolding protein have been postulated in the past, but the present study provides more compelling evidence for such a role. Overall the experiments are well performed and adequately controlled, and the authors conclusions are valid. That being said, there are there are limitations to the study that need to be addressed before this work would be ready for further consideration. Specific issues are listed below.

- 1) Perhaps the most important issue is whether or not AKAP8 has a direct interaction with RNA. This is an important issue as AKAP8/AKAP95 was originally discovered as a kinase scaffolding protein. Subsequent analyses have shown that this protein has a range of binding partners. The authors present Bio-ID proteomic data conforming that AKAP8 interacts with RNA, but to my knowledge it remains an open question as to whether this scaffolding protein does so through

direct interaction with RNA or through secondary association with other RNA-binding proteins. This issue has to be more clearly addressed experientially.

2) The informatics in figures 1d-f are not as informative as they could be. More clarity in this section is needed. This is especially important since an artificial inducible model of EMT was used to implicate AKAP8 in earlier sections of this figure.

3) RNAi depletion of AKAP8 is used as a means to establish a role for this protein in EMT transitions. There are several key controls missing from this section. i) AKAP8 is a predominantly nuclear protein. immunocytochemical analyses should be included in the body of the text showing that gene silencing depletes the nuclear AKAP8 signal with both knockdown probes. These experiments would solidify the findings presented in figure 2c. ii) rescue experiments upon ectopic expression of AKAP8 in the same cell lines as used in figure 2 must be included.

4) Why are the animal studies in figure 2 so variable. Two different AKAP8 knockdown probes apparently give strikingly different results in figure 3c (compare red and green lines). This needs to be explained.

5) The data in figures 5 & 6 are interesting, but again beg the question of whether or no AKAP8 interaction with RNA is really direct or via a secondary binding partner. Mapping of the nucleotide recognition sequences is powerful but needs to be augmented by in vitro mapping studies with AKAP8 to define the reciprocal RNA binding surface on the scaffolding protein. Alternatively (see point 1) the authors may discover that an AKAP8 binding partner interfaces with RNA. These studies would go a long way to provide a mechanistic explanation for the phenomena observed in this study.

6) The authors provide compelling evidence for a nuclear function of AKAP8 and should consolidate this evidence by pointing out that, although discovered as an A-Kinase-Anchoring protein, there is little if any evidence for AKAP8 association with PKA in the nucleus. A simple statement in the discussion would suffice and strengthen the authors arguments.

Reviewer #3:

Remarks to the Author:

This is an interesting study on alternative splicing and its putative role in EST regulation of breast cancer tumor cell lines. Starting from a mass spec identification of 50 hnRNPM interacting proteins in 293 cells – hnRNPM has been known as a factor catalyzing exon skipping – the authors screened these by ectopic expression for effects on splicing using CD44 variant exon minigenes. One of the interacting proteins, AKAP8, inhibited exon sequence excision, shown for both CD44v8 and v5. The authors delineated a yet unknown function of AKAP8. It inhibited EMT (epithelial-to-mesenchymal transition) in a number of cells in culture. Given the role of EMT in promoting migration, it is not surprising that tumor cells modulated for expression of such factors show altered lung colony formation upon tail vein injection into immune-compromized mice. Knocking down AKAP8 in HIM3 cells (derived from triple negative breast cancer, apparently behaving epithelial-like) decreased expression of the epithelial markers E-cadherin, γ -catenin, and occludin, and upon tail vein injection into immune-suppressed mice produced more and larger metastatic lung colonies. The reverse was observed by overexpressing AKAP8 in metastatic MDA-MB-231 derived LM2 cells.

The authors then analyzed splicing by global transcriptome analysis under the influence of AKAP8. shRNA downregulation of AKAP8 in epithelial cells carrying tamoxifen-inducible Twist (an inducer of EMT) reduced E-cadherin expression and caused faster development of EMT. Interestingly, the response differed whether the cells were non-induced by tamoxifen, were in the process of induction, or had fully arrived at the EMT state. This suggests that additional regulatory factors influence the action of AKAP8. A global assessment of alternative splicing was done by deep

sequencing of RNA. A large number of transcripts are indeed alternatively spliced and are influenced by AKAP8. The transcripts were grouped and compared with those elevated in the EST state. The action on CD44v5 and v8 as well on so many other transcripts indicates that AKAP8 influenced splicing in more general terms irrespective of any exon specificity.

The mechanism of AKAP8 action was elaborated. It inhibited the binding of hnRNPM to its intronic sites downstream of CD44 exons v5 and v8 and antagonized hnRNPM dependent exon skipping. AKAP8 downregulation enhanced the hnRNPM binding in epithelial-state cells, but not in twist dependent mesenchymal cells, suggesting additional cell state dependent factors as mentioned above.

After identifying a strong AKAP8 binding motif, one target gene was identified by single end crosslinking and immunoprecipitation: calsyntenin (CLSTN1). An AKAP8 binding site is located at the upstream proximal intron of variable exon 11 of CLSTN1.

As a possible relevance of AKAP8 and its target CLSTN1 for human cancer might be derived from patient data from breast cancer data collections: high AKAP8 levels correlated with better breast cancer patient metastasis-free survival. Similarly correlated alternative splicing of CLSTN1 with breast cancer patient survival.

Comments:

The study is convincing with respect to the molecular mechanism how AKAP8 counteracts exon sequence excision and antagonizes the transition from the epithelial to the mesenchymal phenotype in cell cultures, and that AKAP8 inhibits lung colonization of breast cancer cell lines in immune-suppressed mice. That splice regulation by AKAP8 and its target calsyntenin may be relevant for human patient survival, is plausible by the data derived from the large cohort in the breast cancer data sets.

AKAP8 antagonizes EMT-associated alternative splicing across the transcriptome to maintain an epithelial cell-state. It would be interesting to put these findings into modeling programs and self-learning machinery. Predictions could be generated for the pathways of the enormous splice alterations induced by AKAP8.

Perhaps somewhat naive and overinterpreted are the conclusions that despite the overall massive alteration in splicing the inhibition of exon skipping in CD44 or calsyntenin would be limiting in tumor metastasis. Such conclusions would require more complex animal experimentation. The general inhibition of exon skipping is apparently not detrimental for the cells in culture.

The role of CD44 and its splice variants is more complicated than introduced in the current manuscript, in that not only the tumor cells take advantage from CD44 expression, but all cells of the immune system including those which inhibit or stimulate cancer, depend on CD44 in one way or the other. Experiments with immune-suppressed mice exclude part of this relevant tumor microenvironment.

The selected literature on CD44 is biased in that those references are selected which relate CD44 exon skipping with tumors while the opposite can also be found in the literature.

Abbreviations should be clarified. For Instance in the legends to figures. One can understand by guessing, but the legend to fig. 4 could explain AK8 and hnM. Dito PSI.

Reviewers' comments

Reviewer #1 (Remarks to the Author):

Hu et al. report the identification of AKAP8 as an RNA binding protein that inhibits EMT and breast cancer metastasis through the regulation of alternative splicing. They demonstrate that AKAP8 interacts with hnRNPM and compete with hnRNPM to stimulate exon skipping of CD44. They show that AKAP8 is capable of directly binding to RNA and modulates alternative splicing events. They assess the function of one an AKAP8 splicing target, CLSTN1, in EMT.

Findings from this study contribute to improving our understanding of the regulators of breast cancer metastasis and highlight the role of splicing regulators in EMT and tumor progression. The data presented here is very clear and provides novel insights into the role of AKAP8 in metastasis.

A few questions should be addressed before the manuscript is ready for publication:

MAJOR COMMENTS:

1) Fig.1- Is there a difference in survival in breast tumors with high- vs. low-AKAP8 expression if subdivided by tumor subtype? Additionally, is there a correlation between AKAP8 and HNRMP expression in breast tumors?

Response to reviewer:

We thank the Reviewer for his/her recognition of the importance of our work. According to the Reviewer's suggestion, we stratified breast tumors based on their intrinsic molecular subtype and investigated the relationships between AKAP8 expression and patient survival in each tumor subtype. Our results showed that high AKAP8 expression correlates with better survival in the Luminal A, Luminal B, and HER2 breast cancer subtypes and not in the Basal and Claudin-low subtypes. As the Luminal A, Luminal B, and HER2 tumors are more epithelial and the Basal and Claudin-low tumors are more mesenchymal in nature, these clinical results are in line with our experimental findings that AKAP8 acts in epithelial states for maintaining an epithelial phenotype. We now included these results in Supplementary fig. 1c – 1g.

We also analyzed the relationships between the expression of AKAP8 and hnRNPM and did not find significant correlation at the gene expression level. HnRNPM is an abundant gene expressed in all tumor cells. We speculate that the activity of hnRNPM could be regulated by post-translational modification or by its binding partner during EMT. We have now discussed the relationships of AKAP8 and hnRNPM on Page 12 and Page 22.

2) Fig. 4- The authors nicely demonstrate that increased levels of AKAP8 can affect CD44 splicing in presence of HNRNPM, and reciprocally that decreased AKAP8 levels can have the opposite effect. Given that AKAP8 binds to HRNPM, that AKAP8 also binds to RNA, it remains unclear if AKAP8 can also modulate CD44 splicing in cells lacking HNRNPM, thus demonstrating that it can bind directly, or whether AKAP8 affects splicing by sequestering HNRNPM away from that transcript? This should at least be discussed.

Response to reviewer:

We thank the reviewer for raising this interesting point. Accordingly, we examined the effect of AKAP8 on regulating CD44 splicing in cells that have had hnRNPM depleted by siRNA. We found that

AKAP8 promotes CD44 v8 exon inclusion. Silencing hnRNPM in these AKAP8 expressing cells resulted in a moderate but not significant increase in AKAP8's splicing activity. Moreover, our eCLIP data indicated an AKAP8 binding peak located upstream of the v8 exon from, and we were able to experimentally validate the binding. These results suggest that AKAP8 can function independent of hnRNPM. We have now included the data in Supplementary fig. 4e, 4f, 6d, and a discussion on Page 21.

3) Fig 5. and Fig. 6- The authors demonstrate that AKAP8 affects splicing of a set of target isoforms, and that AKAP8 binds directly to an RNA motif. How many of the spliced isoforms regulated by AKAP8 as detected by RNA-seq (Fig. 5) are also bound by AKAP8 as detected by eCLIP (Fig. 6)? Is there are difference in the position of AKAP8 binding site in exon that are included vs. skipped? Additionally, the authors should provide the location of AKAP8 and HNRNPM binding sites in the CD44 and CLSTN1 transcripts and discuss whether these two proteins are likely to compete for binding to the same sites?

Response to reviewer:

We identified 16 and 20 splice isoforms regulated by AKAP8 that are also bound by AKAP8 in epithelial and mesenchymal cell states, respectively. These events as well as the location where they are bound are detailed in Supplementary Table 3. We did not identify a significant association between binding topology and AKAP8's promotion of inclusion or skipping. We have updated this information in the Results section on Page 16.

In addition, we have added AKAP8 and HNRNPM binding sites in the CD44 and CLSTN1 pre-mRNAs in Supplementary Fig. 6c and Fig. 6f, respectively. We have also added to the discussion regarding their similar and sometimes overlapping binding sites in the discussion on Page 21, 22.

4) The authors demonstrate that AKAP8 and HNRPM both regulate CD44 isoforms using a splicing reporter minigene. Using RNA-seq data from cells with AKAP8 or HNRPM KD have the authors been able to identify other targets that are also co-regulated by both?

Response to reviewer:

Indeed, hnRNPM and AKAP8 do regulate common splicing targets including CD44 and CLSTN1. We extracted splicing events regulated by hnRNPM in our published paper (Harvey et al., 2018, PMID: 30042172) and overlapped them with those regulated by AKAP8. 18 events, including CD44 and CLSTN1, overlapped in the epithelial state while 14 events overlapped in the mesenchymal state. We included this information in the discussion section on Page 21-22 and did not include it in the Results section as we intend to maintain the focus of current manuscript. However, if the reviewer requests, we are happy to include this information in the supplemental data. The results are attached below.

5) The author previously uncovered a co-regulation of HNRPM and ESRP1 splicing targets (Harvey et al. RNA 2018). This begs the question whether ESRP1 or other RBP levels are affected by AKAP8 KD or OE. This should at least be discussed.

Response to reviewer:

We thank the Reviewer to raise this interesting question. We extracted RBPs that are differentially expressed in response to AKAP8 KD from our RNA sequencing data. There are a handful of RBPs that are regulated upon AKAP8 knockdown, however, the levels of hnRNPM and ESRP1 are not altered. Examination of the expression levels of splicing factors that have been implicated in regulating EMT, including RBFOX2, QKI, and RBM47, showed that these factors were not differentially expressed upon AKAP8 knockdown. Additionally, we experimentally examined the expression of these splicing factors (except for RBM47 whose expression is extremely low) in response to AKAP8 knockdown and the results are consistent. We have added these points in Discussion on page 22. The qRT-PCR results are attached below.

6) Fig. 4f, g – The author state that “*hnRNPM’s binding signal was not altered in AKAP8 knockdown cells in the mesenchymal state [...] - suggested that post-translational modification of AKAP8 and/or hnRNPM are involved in contributing to this epithelial cell state-specific activity, an area of research that requires further exploration.*”. Here again, the authors should at least discuss the possibility that these differences are mediated by changes in levels of other hnRNPM binding partners identified in Fig. 1 or levels of other co-regulators, such as ESRP1 identified in Harvey et al. 2018.

Response to reviewer:

We thank the reviewer for this comment. We now added the following statement on Page 12: “These possibilities could include post-translational modification of AKAP8 and/or hnRNPM, changes in levels of other hnRNPM binding partners identified in Fig. 1, and changes in expression of the previously reported splicing regulator ESRP1, which serves as an antagonist of hnRNPM’s splicing activity^{9,10}.”

MINOR COMMENTS:

1) Please italicize gene and transcript names, e.g., *CD44 gene* or *minigene* throughout the manuscript.

Response to reviewer:

We have edited the manuscript now per these specifications.

2) *Fig. 1- The experiments presented in Fig. 1 and associated supplemental data should be described more clearly. For example, what exactly is the assay used for screening candidate splicing factors? Is the assay's read-out fluorescence or CD44 isoform levels as detected by RT-PCR? How were the 29 SFs selected from the 51 candidates identified by BioID? What are the expression levels of the 29 SFs?*

Response to reviewer:

According to the Reviewer's request, we have now provided more detailed descriptions for Fig. 1 and associated supplemental data on page 6. It reads: "From the top 50 hnRNPM-interacting proteins, we selected 29 splicing factors (See Methods for details) and performed a *CD44v8* minigene reporter assay (Supplementary Fig. 1b and Supplementary Table 1). After co-transfection of each of the 29 ORFs with the *CD44v8* minigene reporter to 293 FT cells, we collected RNA and performed qRT-PCR analysis on the levels of *CD44v8* inclusion and skipping. The ratios of inclusion to skipping were then compared." We have also included the rationale of selection on the 29 splicing factors in the Methods section on page 26. It reads: "29 ORFs of splicing factors were obtained as ORF entry clones. They were then cloned into a pLenti6.3 V5/Dest vector to generate the destination clones by the gateway LR reaction (Invitrogen, USA). The 29 splicing factors were selected based on the rank of the unique peptide reads of Mass Spec and the availability of the clones in our cDNA library^{73,74}."

As regard to the expression levels of the 29 SFs, they are all abundantly expressed in 293FT cells according to our RNA-seq data, with TPM higher than 10.

3) *Fig. 1d and Fig. 7g- Please include number of patients in each subclass for survival plots.*

Response to reviewer:

The number of patients in each subclass has been added to the figures.

4) *Fig. 3 - Please provide data from the ctrl plasmid +/- DOX to control for dox-induced changes in Fig. 3g. Please also provide quantification and number of samples for histology analysis shown in Fig. 3j.*

Response to reviewer:

We now included a western blot image in Supplementary Fig. 3b showing AKAP8 protein levels in the ctrl and AKAP8 cDNA overexpressing cells with and without DOX treatment. Only AKAP8 cDNA-expressing cells showed significant upregulation of AKAP8 level after DOX induction.

We also quantified the areas of metastatic lesions of mouse lungs and now included the quantification and number of samples in Fig. 3j.

5) *Fig. 4 -The difference in splicing ratio detected by QPCR vs. semi-quantitative PCR are following the same trend, but the values are quite difficult to interpret. Please clarify, or normalize the QPCR value to total transcript expression and plot a PSI value between 0-100%.*

Response to reviewer:

We performed qPCR analysis using isoform specific primers to detect the inclusion and skipping isoforms, respectively. We then took the ratio of the inclusion to skipping. Because PCR from two primer sets can give rise to different amplification efficiency, we avoid taking the sum of the values of two PCR

products as a denominator for calculating the percent spliced in (PSI). As for the semi-qPCR experiments, we used the same primer set to obtain both the inclusion and skipping products. We therefore took the sum of the two as the total and calculated the PSI value. We could on the other hand convert these PSI values to the ratio of inclusion to skipping. Using Fig. 4a as an example, the relative inc/skip ratios to the control are: 1, 0.43, 0.83, 1.12, 1.6, in line with the results shown in Fig. 4a top panel. We now modified the method section to indicate the rationale on performing the ratios of inc/skip in qPCR analysis on Page 27.

6) *Supplementary Fig. 5a- The figure does not really answer the question how many of the changes in spliced isoform are also associated with changes in gene expression levels. Could the author provide a Venn diagram with overlapping gene IDs from the differentially spliced isoform and differentially expressed gene lists?*

Response to reviewer:

Thank you. We have now provided this venn diagram in Supplementary Fig. 5b.

7) *Fig. 5d,e - Where are the EMT AS events derived from? Please add description and reference to dataset.*

Response to reviewer:

We performed RNA sequencing in epithelial and mesenchymal HMLE/Twist-ER control and AKAP8 KD cells. The EMT AS events were derived from the control epithelial and mesenchymal cells. We obtained a set of cassette exon skipped events (SEs) whose Δ PSI values were altered during EMT (threshold setting at FDR < 0.05, $|\Delta$ PSI \geq 0.1). This has been clarified in the Results section on Page 13: "Since SEs represent the vast majority of AKAP8 regulated alternative splicing events, we next overlapped AKAP8 regulated SEs with those regulated during EMT. The EMT regulated splicing events were obtained from the differential splicing events in control HMLE/Twist-ER cells that are in either the epithelial state or the mesenchymal state." We thank the reviewer for raising this point. It helps us present the results in a clearer manner.

8) *Fig. 6c- Please scale panels to the same values to allow comparison.*

Response to reviewer:

We have now scaled the panels to the same Y-axis to allow direct comparison.

9) *Fig. 6f- Please show IP signal as fold enrichment over input*

Response to reviewer:

Tracks now show IP/Input Signal in 5-nucleotide bins.

10) *Fig. 7f- Here the authors use vimentin levels as a mesenchymal marker. What is vimentin expression in cell lines used throughout the paper? Why not use E-cadherin as all other figures?*

Response to reviewer:

Vimentin is one of the common markers of the mesenchymal state. We tried to use the vimentin antibody from ThermoFisher (catalogue #. MS-129-P) to perform western blot analysis using lysates from cell lines, but the signals were extremely weak. We therefore used another mesenchymal marker N-cadherin, which was able to detect in the HMLE-Twist-ER cell lines we used in the study. The epithelial marker E-cadherin is found to localize at cell surface or in cytoplasm. When located in the cytoplasm, E-cadherin no longer functions as an epithelial cell adherens junction protein. We performed the correlation analysis between E-cadherin and CLSTN1 PSI and did not find a significant correlation. As the E-cadherin mRNA levels do not reflect the cell surface E-cadherin, we suspected that this could contribute to the lack of significant correlation.

11) Fig. 7g- Please indicate the cut-off used for defining tumors as low vs. high CLSTN1 PSI.

Response to reviewer:

The CLSTN1 PSI high and low groups were determined by unbiased k-means clustering, therefore no cutoffs were used to distinguish the groups. This information is included in the Methods section. The average PSI for the “high” group is 0.49. The average PSI for the “low” group is 0.15. This information is now included in the legend of Fig. 7g.

Reviewer #2 (Remarks to the Author)

This is a well written and articulated article that documents observations that the A-Kinase Anchoring protein AKAP8/AKAP95 participates as a novel metastasis suppressor by antagonizing EMT responsive alternative splicing of RNA. Similar roles for this scaffolding protein have been postulated in the past, but the present study provides more compelling evidence for such a role. Overall the experiments are well performed and adequately controlled, and the authors conclusions are valid. That being said, there are there are limitations to the study that need to be addressed before this work would be ready for further consideration. Specific issues are listed below.

1) Perhaps the most important issue is whether or not AKAP8 has a direct interaction with RNA. This is an important issue as AKAP8/AKAP95 was originally discovered as a kinase scaffolding protein. Subsequent analyses have shown that this protein has a range of binding partners. The authors present Bio-ID proteomic data conforming that AKAP8 interacts with RNA, but to my knowledge it remains an open question as to whether this scaffolding protein does so through direct interaction with RNA or through secondary association with other RNA-binding proteins. This issue has to be more clearly addressed experientially.

Response to reviewer:

We thank the Reviewer’s positive comments. The contention that AKAP8 directly binds RNA is supported by the presence of stringent AKAP8 binding peaks identified by our eCLIP assay in close proximity to AKAP8 regulated splicing events. The eCLIP assay involves crosslinking between protein and RNA nucleotides that are in angstrom-level proximity followed by stringent washing and immunoprecipitation. Thus, eCLIP results directly indicate the binding sites of an RBP. Moreover, to ensure that our AKAP8 binding peaks are significantly enriched relative to background RNA levels, we used size-matched input RNA sequencing. ECLIP peaks and experimental validation of AKAP8 binding to CD44 and CLSTN1 at sites identified by eCLIP (Fig. 6f, Supplementary Fig. 6c, 6d, Fig 7a) further provided strong evidence that AKAP8 directly binds to RNA.

2) *The informatics in figures 1d-f are not as informative as they could be. More clarity in this section is needed. This is especially important since an artificial inducible model of EMT was used to implicate AKAP8 in earlier sections of this figure.*

Response to reviewer:

We thank the reviewer for raising this point. We now included more detailed information in the text and figure legend. On Page 7, it reads: "As our goal was to identify splicing factors that regulate EMT and cancer metastasis, we used a breast cancer database and examined whether the expression of any of the above identified five splicing factors predicts tumor metastasis and patient survival. AKAP8 showed the most significant correlation with metastasis and patient survival (Fig. 1d). AKAP8 expression positively correlates with distal metastasis free survival in a cohort of 327 published breast cancer samples analyzed by microarray⁴⁹. The positive correlation of AKAP8 expression and metastasis-free survival is congruent with our experimental findings that AKAP8 promotes CD44v8 inclusion and inhibits CD44s production, the isoform that promotes EMT and tumor metastasis^{8,30,33}. Further analysis of the METABRIC breast cancer dataset showed that AKAP8 expression positively correlates with overall survival (Fig. 1e), most significantly in Luminal A, Luminal B, and HER2+ subtypes (Supplementary Fig. 1c – 1g). Analysis of the AKAP8 expression levels in different subtypes of breast cancer revealed that AKAP8 expression is highest in the Luminal A subtype, the least aggressive breast cancer subtype that is epithelial in nature, and AKAP8 expression is significantly lower in the Claudin Low and Basal subtypes, which are more aggressive and are mesenchymal in nature (Fig. 1f and Supplementary Fig. 1h). Similarly, AKAP8 is highly expressed in the ER-positive breast tumors compared to the ER-negative breast tumors (Supplementary Fig. 1i). These results associate AKAP8 expression with the epithelial phenotype in breast cancer and show that loss of AKAP8 is a characteristic of poor survival, prompting us to explore the role of AKAP8 more closely using in vitro models of EMT and in vivo models of breast cancer metastasis."

3) *RNAi depletion of AKAP8 is used as a means to establish a role for this protein in EMT transitions. There are several key controls missing from this section. i) AKAP8 is a predominantly nuclear protein. immunocytochemical analyses should be included in the body of the text showing that gene silencing depletes the nuclear AKAP8 signal with both knockdown probes. These experiments would solidify the findings presented in figure 2c. ii) rescue experiments upon ectopic expression of AKAP8 in the same cell lines as used in figure 2 must be included.*

Response to reviewer:

According to the Reviewer's request, we performed AKAP8 immunofluorescent analysis. Our results showed that AKAP8 signals were located in the nucleus and were drastically depleted in both AKAP8 KD cells. The result is now included in Supplementary Fig. 2b.

We have also re-expressed the AKAP8 cDNA in the HCT116 cells that express shAKAP8, the cell line that was used Fig. 2d, 2e. We now show that re-expressing AKAP8 restored the decreased expression of epithelial markers in shAKAP8 cells in Supplementary Fig. 2c. In addition, we showed in Fig. 2a – 2c that silencing AKAP8 by two independent AKAP8 shRNAs resulted in an acceleration of EMT. Furthermore, we showed in Supplementary Fig. 3c that ectopic expression of AKAP8 in LM2 cells resulted in an increase in the epithelial marker expression and a decrease in the mesenchymal marker exoression. Thus, these gain-and-loss of function experiments demonstrate that AKAP8 inhibits EMT.

4) *Why are the animal studies in figure 2 so variable. Two different AKAP8 knockdown probes apparently give strikingly different results in figure 3c (compare red and green lines). This needs to be explained.*

Response to reviewer:

We thank the reviewer for raising this concern. Accumulating evidence has suggested that a hybrid epithelial/mesenchymal (E/M) state has an advantage for tumor cells to survive in the circulation. The ability of the hybrid E/M cells to metastasize is greater than those that are in a mesenchymal state (References 51 – 54 in manuscript).

In our study, the shAKAP8-2 expressing HIM3 cells showed a decrease in epithelial markers as compared to control cells, but to a less extent as compared to the shAKAP8-1 expressing cells, which showed a more complete loss of epithelial markers (Fig. 3a). We speculate that the shAKAP8-2 HIM3 cells could be at a hybrid E/M state that are more prone to metastasis than the mesenchymal shAKAP8-2 HIM3 cells. We now included this speculation in the Results on Page 10. It reads: “We also noticed that the AKAP8 KD-2 cells showed greater metastasis ability than the KD-1 cells, although the KD-2 shRNA was less potent in promoting the EMT phenotype than the KD-1 shRNA. This observation could be in support of the growing evidence that cells in a hybrid epithelial/mesenchymal (E/M) state tend to have a greater advantage in establishing metastatic lesions⁵¹⁻⁵⁴.”

5) The data in figures 5 & 6 are interesting, but again beg the question of whether or no AKAP8 interaction with RNA is really direct or via a secondary binding partner. Mapping of the nucleotide recognition sequences is powerful but needs to be augmented by in vitro mapping studies with AKAP8 to define the reciprocal RNA binding surface on the scaffolding protein. Alternatively (see point 1) the authors may discover that an AKAP8 binding partner interfaces with RNA. These studies would go a long way to provide a mechanistic explanation for the phenomena observed in this study.

Response to reviewer:

Please see our response to Comment #1 on the evidence that AKAP8 directly binds to RNA.

The reviewer is correct that not all alternative splicing events regulated by AKAP8 are through direct AKAP8 binding. Events without bona-fide binding sites represent indirectly regulated events, either through interaction with other RNA binding proteins or secondary effects such as alteration of expression of splicing factors. In a similar manner like many other splicing factors, including RBFOX and hnRNPM, AKAP8 directly and indirectly regulates alternative splicing through both direct binding to the targets or indirect effects on the target splicing.

Regarding the mapping of the AKAP8 RNA binding surface on the protein, there was a previously published paper exploring this question extensively (Hu et al., 2016, PMID: 27824034). They identified that the AKAP8 N-terminal region is essential for AKAP8 interacting with other splicing factors, including hnRNPM, while the two zinc finger domains at the C-terminal are critical for its binding to pre-mRNA. These results indicate that AKAP8 binds to RNA directly through the zinc fingers or indirectly through the N-terminal domain association with other splicing factors. We have included these lines of information in the Discussion section on Page 21.

6) The authors provide compelling evidence for a nuclear function of AKAP8 and should consolidate this evidence by pointing out that, although discovered as an A-Kinase-Anchoring protein, there is little if any evidence for AKAP8 association with PKA in the nucleus. A simple statement in the discussion would suffice and strengthen the authors arguments.

Response to reviewer:

We thank the reviewer to point this out. We did test the relationship between AKAP8 and PKA in affecting CD44v8 splicing. We used existing PKA activators Forskolin and 6-MBC, and inhibitors H-89 and Ht-31. As can be seen in the attached figure, these PKA chemical modulators did not show obvious effects on AKAP8-mediated splicing. We now added this point in the discussion section on Page 20. It reads: “In this study, however, we did not observe effects of PKA signaling on AKAP8’s splicing activity (data not shown)”.

Reviewer #3 (Remarks to the Author)

This is an interesting study on alternative splicing and its putative role in EST regulation of breast cancer tumor cell lines. Starting from a mass spec identification of 50 hnRNPM interacting proteins in 293 cells – hnRNPM has been known as a factor catalyzing exon skipping – the authors screened these by ectopic expression for effects on splicing using CD44 variant exon minigenes. One of the interacting proteins, AKAP8, inhibited exon sequence excision, shown for both CD44v8 and v5. The authors delineated a yet unknown function of AKAP8. It inhibited EMT (epithelial-to-mesenchymal transition) in a number of cells in culture. Given the role of EMT in promoting migration, it is not surprising that tumor cells modulated for expression of such factors show altered lung colony formation upon tail vein injection into immune-compromized mice. Knocking down AKAP8 in HIM3 cells (derived from triple negative breast cancer, apparently behaving epithelial-like) decreased expression of the epithelial markers E-cadherin, γ -catenin, and occludin, and upon tail vein injection into immune-suppressed mice produced more and larger metastatic lung colonies. The reverse was observed by overexpressing AKAP8 in metastatic MDA-MB-231 derived LM2 cells.

The authors then analyzed splicing by global transcriptome analysis under the influence of AKAP8. shRNA downregulation of AKAP8 in epithelial cells carrying tamoxifen-inducible Twist (an inducer of EMT) reduced E-cadherin expression and caused faster development of EMT. Interestingly, the response differed whether the cells were non-induced by tamoxifen, were in the process of induction, or had fully arrived at the EMT state. This suggests that additional regulatory factors influence the action of AKAP8. A global assessment of alternative splicing was done by deep sequencing of RNA. A large number of transcripts are indeed alternatively spliced and are influenced by AKAP8. The transcripts were grouped and compared with those elevated in the EST state. The action on CD44v5 and v8 as well on so many other transcripts indicates that AKAP8 influenced splicing in more general terms irrespective of any exon specificity.

The mechanism of AKAP8 action was elaborated. It inhibited the binding of hnRNPM to its intronic sites downstream of CD44 exons v5 and v8 and antagonized hnRNPM dependent exon skipping. AKAP8 downregulation enhanced the hnRNPM binding in epithelial-state cells, but not in twist dependent mesenchymal cells, suggesting additional cell state dependent factors as mentioned above.

After identifying a strong AKAP8 binding motif, one target gene was identified by single end crosslinking and immunoprecipitation: calsyntenin (CLSTN1). An AKAP8 binding site is located at the upstream proximal intron of variable exon 11 of CLSTN1.

As a possible relevance of AKAP8 and its target CLSTN1 for human cancer might be derived from patient data from breast cancer data collections: high AKAP8 levels correlated with better breast cancer patient metastasis-free survival. Similarly correlated alternative splicing of CLSTN1 with breast cancer patient survival.

The study is convincing with respect to the molecular mechanism how AKAP8 counteracts exon sequence excision and antagonizes the transition from the epithelial to the mesenchymal phenotype in cell cultures, and that AKAP8 inhibits lung colonization of breast cancer cell lines in immune-suppressed mice. That splice regulation by AKAP8 and its target calsyntenin may be relevant for human patient survival, is plausible by the data derived from the large cohort in the breast cancer data sets.

AKAP8 antagonizes EMT-associated alternative splicing across the transcriptome to maintain an epithelial cell-state. It would be interesting to put these findings into modeling programs and self-learning machinery. Predictions could be generated for the pathways of the enormous splice alterations induced by AKAP8.

Response to reviewer:

We thank the Reviewer for his/her recognition of our work. We agree with the reviewer that the modeling program using machine learning is powerful for predictions of pathways and splicing alterations induced by AKAP8. In order to perform such machine learning analysis, it requires a large quantity of RNA sequencing dataset (≥ 100 samples). While this is not feasible in our study, we have performed analyses in Figure 1 that implicate AKAP8 expression in breast cancer patient outcomes including survival and metastasis. Through GSEA analysis we also showed that AKAP8 depletion causes upregulation of mesenchymal signature genes. We agree with the reviewer that it would be interesting to investigate a larger role of the AKAP8 splicing regulon in predication of survival or cancer phenotypes, and this offers interesting future directions.

Perhaps somewhat naive and overinterpreted are the conclusions that despite the overall massive alteration in splicing the inhibition of exon skipping in CD44 or calsyntenin would be limiting in tumor metastasis. Such conclusions would require more complex animal experimentation. The general inhibition of exon skipping is apparently not detrimental for the cells in culture.

Response to reviewer:

According to the Reviewer's comments, we have carefully gone through the manuscript and edited the places that might have overinterpretation. Previously published evidence from our lab revealed a critical role for CD44 exon skipping in *in vivo* mouse models of breast cancer recurrence or metastasis (Brown et al., 2011; Zhao et al., 2016). We agree with the reviewer's comment that we have not conducted the requisite experiments to functionally implicate CLSTN1 exon inclusion in metastasis. However, we have provided evidence for AKAP8's role in suppressing metastasis. We have adjusted our conclusions regarding CLSTN1 alternative splicing on Page 24. It reads: "Future studies on the role of

CLSTN1 splice isoforms in breast cancer metastasis will be necessary to determine if this splice isoform switch is functionally important for metastasis, as we previously uncovered for the CD44 splice isoform switch^{8,30}.”

The role of CD44 and its splice variants is more complicated than introduced in the current manuscript, in that not only the tumor cells take advantage from CD44 expression, but all cells of the immune system including those which inhibit or stimulate cancer, depend on CD44 in one way or the other. Experiments with immune-suppressed mice exclude part of this relevant tumor microenvironment.

Response to reviewer:

Indeed, the role of CD44 and its splice isoforms is very complicated. As the reviewer noted, CD44 expression may affect the immune cells, which may also play an important role in regulating tumorigenesis. We have now included in the discussion about this point on Page 21. It reads: “In this study, we have provided evidence on the role of AKAP8 in suppressing metastasis using immune-compromised mice. As tumor microenvironment plays important roles in both inhibiting and promoting tumor metastasis, future validation on these results in immune competent mice will be needed to better understand the role of the RNA metabolism in metastasis.”

The selected literature on CD44 is biased in that those references are selected which relate CD44 exon skipping with tumors while the opposite can also be found in the literature.

Response to reviewer:

We thank the reviewer for noting the complexity of CD44 isoform function. We have now added the following to the introduction to discuss both CD44 exon skipping and inclusion on Page 4. It reads: “The functional connection of alternative splicing to EMT and cancer metastasis was established through the study of the *CD44* gene, whose alternative splicing generates two families of proteins, known as CD44v and CD44s. After our initial discovery that CD44 isoform switching is essential for EMT⁸, many studies have also reported that epithelial cells that predominantly express CD44v demand an isoform switch to CD44s in order for cells to undergo EMT and for cancer cells to metastasize¹⁹⁻³⁰. While the CD44s splice isoform has recognized roles in promoting cancer cell survival, tissue invasion, cancer stem cell (CSC) traits, and ultimately metastasis^{27,29-33}, CD44v was shown to promote gastric CSC function, and CD44v regulates ROS defense for tumor growth in gastric cancer³⁴⁻³⁸. These different roles of CD44 isoforms could be due to tissue/cancer specificity and suggest the importance in studying splice isoforms in the context of cancer phenotypes.”

Abbreviations should be clarified. For Instance in the legends to figures. One can understand by guessing, but the legend to fig. 4 could explain AK8 and hnM. Dito PSI.

Response to reviewer:

We have now indicated in the figure legends the abbreviations used.

We thank the Reviewers for their critical comments. Addressing these points improves this manuscript. We have reviewed the revised manuscript to make sure that it meets the specifications of *Nature Communications*. We hope that you find this revised manuscript satisfactory for publication.

Reviewers' Comments:

Reviewer #1:

Remarks to the Author:

The authors have addressed all our comments. This study should be published.

Reviewer #2:

Remarks to the Author:

The authors present impressive data regarding control of metastatic transition from epithelial cells. The revised manuscript is much improved. Most of the reviewer points are addressed, but a few minor issues need to be resolved before this article can be considered complete .

1) The binding motif data for akap8 in this paper do not match the authors' data from nat comm 2016 paper. And there is no mention of the discrepancy in the discussion.

2) The authors show that akap8 binds pre-mRNA at defined motifs and impacts splicing. It can also bind hnRNPM and disrupt hnRNPM's ability to impact splicing. Details of this mechanism are lacking. For instance, the previous nat comm paper (2016) from these authors used truncation mutants to show different binding domains of akap8, and then again to show truncation effects on splicing activity of akap8. Since these reagents exist then, I question why they didn't look at akap8 truncation effects on hnRNPM splicing events. Any additional data on this topic would help readers to understand how the two binding activities of akap8 cooperate to impact CD44 splicing.

3) Proximity-biotinylation/MS was recently used to show that nuclear isoforms of akap18 associate with RNA splicing complexes. Citing this PNAS article is important.

Reviewer #3:

Remarks to the Author:

This is a review of the re-submitted and improved manuscript by Hu et al. Previous reviewers' comments have been extensively answered and the manuscript adjusted.

(1) The experimental data on alternative splice regulation are solid and convincing. In my understanding, the responses to the criticisms of figure details and presentations are satisfying.

(2) The question of overinterpretation: The patient survival data from the two data sets support the conclusion that AKAP8 and CLSTN1 splicing are involved in some step in breast cancer progression. The experiments in the paper show lung colonization after tail vein injection of tumor cell lines derived from patients with aggressive breast cancer. These cells possess already most of the properties required for tumor progression. The manipulation experiments only show that alternative splice regulation improves the fate of the cells after injection and in the invasion/exit through the endothelial barrier. To equal this with promotion or suppression of metastasis, is not supported. Based on the experiments shown I would be more modest in my claims.

(3) Simplification of the roles of CD44 in cancer: The authors responded correctly that their experiments ignored the possible effects of the microenvironment on tumor behavior. By a single fast search for CD44v in cancer brings numerous citations for roles that are opposite to those described and discussed in the current paper. To go into a discussion of these other reports, to claim at all that the alternative splice regulation of CD44 is causal for metastasis, could well be left aside – because the manuscript has a different focus: well done data on splice regulation.

REVIEWERS' COMMENTS:

Reviewer #1 (Remarks to the Author):

The authors have addressed all our comments. This study should be published.

Reviewer #2 (Remarks to the Author):

The authors present impressive data regarding control of metastatic transition from epithelial cells. The revised manuscript is much improved. Most of the reviewer points are addressed, but a few minor issues need to be resolved before this article can be considered complete.

1) The binding motif data for akap8 in this paper do not match the authors' data from Nat Commun 2016 paper. And there is no mention of the discrepancy in the discussion.

Response to reviewer:

A discussion regarding the differences between the binding motifs that were derived from our study and published report (*Nat Commun* 2016) were included on page 20.

2) The authors show that akap8 binds pre-mRNA at defined motifs and impacts splicing. It can also bind hnRNPM and disrupt hnRNPM's ability to impact splicing. Details of this mechanism are lacking. For instance, the previous Nat Commun paper (2016) from these authors used truncation mutants to show different binding domains of akap8, and then again to show truncation effects on splicing activity of akap8. Since these reagents exist then, I question why they didn't look at akap8 truncation effects on hnRNPM splicing events. Any additional data on this topic would help readers to understand how the two binding activities of akap8 cooperate to impact CD44 splicing.

Response to reviewer:

We thank the reviewer for this suggestion. We also thank the reviewer's recognition that "The authors show that akap8 binds pre-mRNA at defined motifs and impacts splicing. It can also bind hnRNPM and disrupt hnRNPM's ability to impact splicing." Detailed mechanisms using a series of truncations of AKAP8 is beyond the scope of current study and will be investigated in the future.

3) Proximity-biotinylation/MS was recently used to show that nuclear isoforms of akap18 associate with RNA splicing complexes. Citing this PNAS article is important.

Response to reviewer:

Thank you. This reference is now added.

Reviewer #3 (Remarks to the Author):

This is a review of the re-submitted and improved manuscript by Hu et al. Previous reviewers' comments have been extensively answered and the manuscript adjusted.

(1) The experimental data on alternative splice regulation are solid and convincing. In my

understanding, the responses to the criticisms of figure details and presentations are satisfying.

Response to reviewer:

We thank the Reviewer's comments.

(2) The question of overinterpretation: The patient survival data from the two data sets support the conclusion that AKAP8 and CLSTN1 splicing are involved in some step in breast cancer progression. The experiments in the paper show lung colonization after tail vein injection of tumor cell lines derived from patients with aggressive breast cancer. These cells possess already most of the properties required for tumor progression. The manipulation experiments only show that alternative splice regulation improves the fate of the cells after injection and in the invasion/exit through the endothelial barrier. To equal this with promotion or suppression of metastasis, is not supported. Based on the experiments shown I would be more modest in my claims.

Response to reviewer:

We accept the reviewer's comment that *in vivo* metastasis investigations focus more on extravasation and colonization of the metastatic cells as opposed to metastasis from a primary site. However, these steps are critical components of metastasis. Given that AKAP8 depletion does not affect cell proliferation, we feel justified in concluding that AKAP8 suppresses metastasis. Future experiments may be performed on investigation of AKAP8 on tumor metastasis from primary tumors.

(3) Simplification of the roles of CD44 in cancer: The authors responded correctly that their experiments ignored the possible effects of the microenvironment on tumor behavior. By a single fast search for CD44v in cancer brings numerous citations for roles that are opposite to those described and discussed in the current paper. To go into a discussion of these other reports, to claim at all that the alternative splice regulation of CD44 is causal for metastasis, could well be left aside – because the manuscript has a different focus: well done data on splice regulation.

Response to reviewer:

We agree with the reviewer that the role of CD44 splice isoforms in cancer is not the focus of this study, and therefore removed this discussion accordingly.